# Amynthas corticis genome reveals molecular mechanisms behind global distribution

Xing Wang[1,2,9✉], Yi Zhang[3,9], Yufeng Zhang[4,9], Mingming Kang[5,9], Yuanbo Li[1,2], Samuel W. James [6], Yang Yang[7], Yanmeng Bi[8], Hao Jiang[1], Yi Zhao [1✉] & Zhenjun Sun [1,2✉]

Earthworms (Annelida: Crassiclitellata) are widely distributed around the world due to their ancient origination as well as adaptation and invasion after introduction into new habitats over the past few centuries. Herein, we report a 1.2 Gb complete genome assembly of the earthworm Amynthas corticis based on a strategy combining third-generation long-read sequencing and Hi-C mapping. A total of 29,256 protein-coding genes are annotated in this genome. Analysis of resequencing data indicates that this earthworm is a triploid species. Furthermore, gene family evolution analysis shows that comprehensive expansion of gene families in the Amynthas corticis genome has produced more defensive functions compared with other species in Annelida. Quantitative proteomic iTRAQ analysis shows that expression of 147 proteins changed in the body of Amynthas corticis and 16 S rDNA sequencing shows that abundance of 28 microorganisms changed in the gut of Amynthas corticis when the earthworm was incubated with pathogenic Escherichia coli O157:H7. Our genome assembly provides abundant and valuable resources for the earthworm research community, serving as a first step toward uncovering the mysteries of this species, and may provide molecular level indicators of its powerful defensive functions, adaptation to complex environments and invasion ability.

[1] College of Resources and Environmental Sciences, China Agricultural University, Beijing, China. [2] Beijing Key Laboratory of Biodiversity and Organic Farming, Beijing, China. [3] School of Environmental Science & Engineering, Southern University of Science and Technology, Shenzhen, Guangdong, China. [4] Hebei Key Laboratory of Animal Diversity, Langfang Normal University, Langfang, Hebei, China. [5] College of Biological Sciences, China Agricultural University, Beijing, China. [6] Maharishi University of Management, Fairfield, IA, USA. [7] Beijing Gencode Diagnostics Laboratory, Beijing, China. [8] School of Environmental and Municipal Engineering, Tianjin Chengjian University, Tianjin, China. [9] These authors contributed equally: Xing Wang, Yi Zhang, Yufeng Zhang, Mingming Kang. ✉email: swwangxing@cau.edu.cn; zhaoyis@163.com; sun108@cau.edu.cn

E arthworms are clitellate annelids distributed in virtually all terrestrial habitats except deserts and icecaps. Recently, a research focus on diversity of earthworm at the global scale showed that this soil and sediment taxon has more local richness and abundance in temperate zones, but more beta diversity across different locations in tropical zones[1]. More interestingly, this meta-analysis also showed that climate is the key factor affecting earthworm distributions, regardless of soil properties and vegetation cover[1]. Soil properties and vegetation represent diverse environments, which means different challenges to species, especially for underground ones. Thus, it could be proposed that there must be unique and robust molecular mechanisms in any earthworm species helping it to exist in adverse and complex environments.

Earthworms have been recognized as a group with profound ecological and economic impacts on soil[2,3] and are considered "soil engineers". On the other hand, some earthworms are also invasive species with important effects on soil ecosystems[4–6], including reduced local diversity of native species[7]. Potential contributions to invasive ability are polyploidy and parthenogenetic reproduction, as has been identified in earthworm species of the family Lumbricidae[8–11]. Polyploidy provides more genomic materials for evolving or expressing novel phenotypes[12,13], while parthenogenetic reproduction ensures the stability of those phenotypes[14], which is beneficial when they are well-matched to a new environment.

*Amynthas corticis* (Kinberg, 1867), is in the Megascolecidae, which has a primarily Asian-Australasian distribution. With physical dispersal, *A. corticis* has become a cosmopolitan invasive species with an east Asian origin. It has been suggested that the invasive ability of *A. corticis* is due to its great adaptability and polyploid or parthenogenetic reproduction[15]. We collected *A. corticis* from the campus of China Agricultural University in July 2017 (Supplementary Fig. 1a–b, Supplementary Fig. 2). *A. corticis* is a common species widely distributed in grassland of parks and schools. The length, width, and body segment number of *A. corticis* are 124 mm, 3.5 mm, and 105. One pair of male pores, which are apart from each other with one fourth of body circumference, and one single female pore locate in venter of the eighteenth and fourteenth body segment, respectively. Three small and circular genital papillae are present around each male pore (Supplementary Fig. 1c). Four pairs of spermathecae locate in venter of intersegments (5/6, 6/7, 7/8, and 8/9) with ovoid ampulla, straight stalk, and blunt ovoid diverticulum (Supplementary Fig 1d). Prostate gland of *A. corticis* is rudimentary to a small duct located in the eighteenth body segment, which is a phenotype concerned parthenogenesis.

To reveal the whole picture of molecular mechanisms behind the ecological plasticity and adaptation of *A. corticis*, we sequenced its genome and generated a complete 1.2 Gb assembly[16], including 42 chromosome-level scaffolds with N50 length of 31 Mb (Table 1 and Supplementary Fig. 3) and annotated a total of 29,256 protein-coding genes in the genome (Supplementary data file 1). Beside protein-coding genes, we also identified repeat sequences, noncoding genes and SNPs and displayed them along with 42 chromosome-level scaffolds (Fig. 1, Supplementary data file 2,3). Then, the evaluation for genome completeness based on BUSCO showed that 91.2% of BUSCOs were complete (including 68.6% single-copy ones and 22.6% duplicated ones), while average number of orthologs per core genes and percentage of detected core genes that have more than 1 ortholog were 1.3 and 24.78, respectively, suggesting well quality of the assembled genome. In addition to the genome, we also performed iTRAQ-based quantitative proteomics analysis to detect differentially expressed proteins of *A. corticis* and sequenced 16 S rDNA from its intestinal tract after treatment

**Table 1 Summary of the assembled genome of *Amynthas cortices*.**

| Assembly | Contigs | Scaffolds |
|---|---|---|
| Total length (bp) | 1,190,923,171 | 1,285,148,884 |
| Number of sequences | 16,882 | 1042 |
| Length of the longest sequence (bp) | 1,225,102 | 55,651,430 |
| N50 length (bp) | 117,165 | 31,950,757 |
| N90 length (bp) | 33,690 | 21,414,444 |
| GC content (%) | 40.34 | 40.34 |
| N content (%) | 0 | 0.26 |
| Number of gaps | 0 | 12,771 |
| Length of gaps (bp) | 0 | 3,283,278 |

with a pathogenic *Escherichia coli* stress O157:H7[17,18] to reveal how this earthworm genome functions when facing stresses. Last, we systematically profiled copies of several categories of well determined defensive genes in the genome, including lysins[19–21], antimicrobial proteins[22–24], Toll-like receptors[25,26], proteins involved in response to oxidative stress[27,28] or metal stress[20], detoxification proteins[27,29], and heat shock protein[30], and found their expansion trend in the genome as well as high diversity in the population of *A. corticis*.

## Results

***Amynthas corticis* is triploid.** To evaluate the ploidy of the genome of *A. corticis*, we called genome-wide SNPs and constructed the coverage distribution of heterozygous k-mer based on resequencing data of *A. corticis* individuals. First of all, the density distribution of allele frequency related to SNPs on biallelic loci shows that there are two peaks near the frequency of 1/3 and 2/3 (Fig. 2a), suggesting most biallelic loci have three copies in the genome of *A. corticis*. In addition, the result of statistical analysis based on the density distribution of allele frequency related to SNPs on biallelic loci shows that the triploid hypothesis possesses the lowest delta log-likelihood[31,32] (Fig. 2b), suggesting the genome of *A. corticis* is most probably triploid. Furthermore, the coverage distribution related to 88% of heterozygous k-mer pairs is concentrated at 1/3 for the normalized coverage of minor k-mer and 3n for the total coverage of k-mer pairs[33] (Fig. 2c), also indicating the genome of *A. corticis* is triploid. Lastly, karyotype analysis of four *A. corticis* individuals shows that chromosome number in each individual is larger than 120 (Fig. 2d, e). Given that we have assembled 42 chromosome-level scaffolds, the number of chromosomes indicates that *A. corticis* is triploid. Thus, multiple independent proofs support our conclusion that the genome of *A. corticis* is triploid. Intriguingly, we found the genome of earthworm *Eisenia fetida*[34] is diploid (Supplementary Fig. 15) with the same analysis, suggesting diversity among earthworms.

**Abundant gene content related to defensive reactions.** Compared to other annelids, *Capitella teleta* and *Helobdella robusta*[35], the genome size of earthworm *A. corticis* was larger than that by 3.87 and 5.49 fold, respectively. This phenomenon suggested that there might be comprehensive expansion of gene families happened in the genome of *A. corticis* after divergence from the most recent common ancestor (MRCA) of annelids. Previous studies have demonstrated that expansion of sodium-potassium pump alpha-subunit and several other gene families in leech happened at the same time as its transition from marine to freshwater habitats and might play important roles in diversification of annelids into freshwater habitats[36,37]. To reveal how expansion of gene families affected earthworm *A. corticis*, we constructed gene

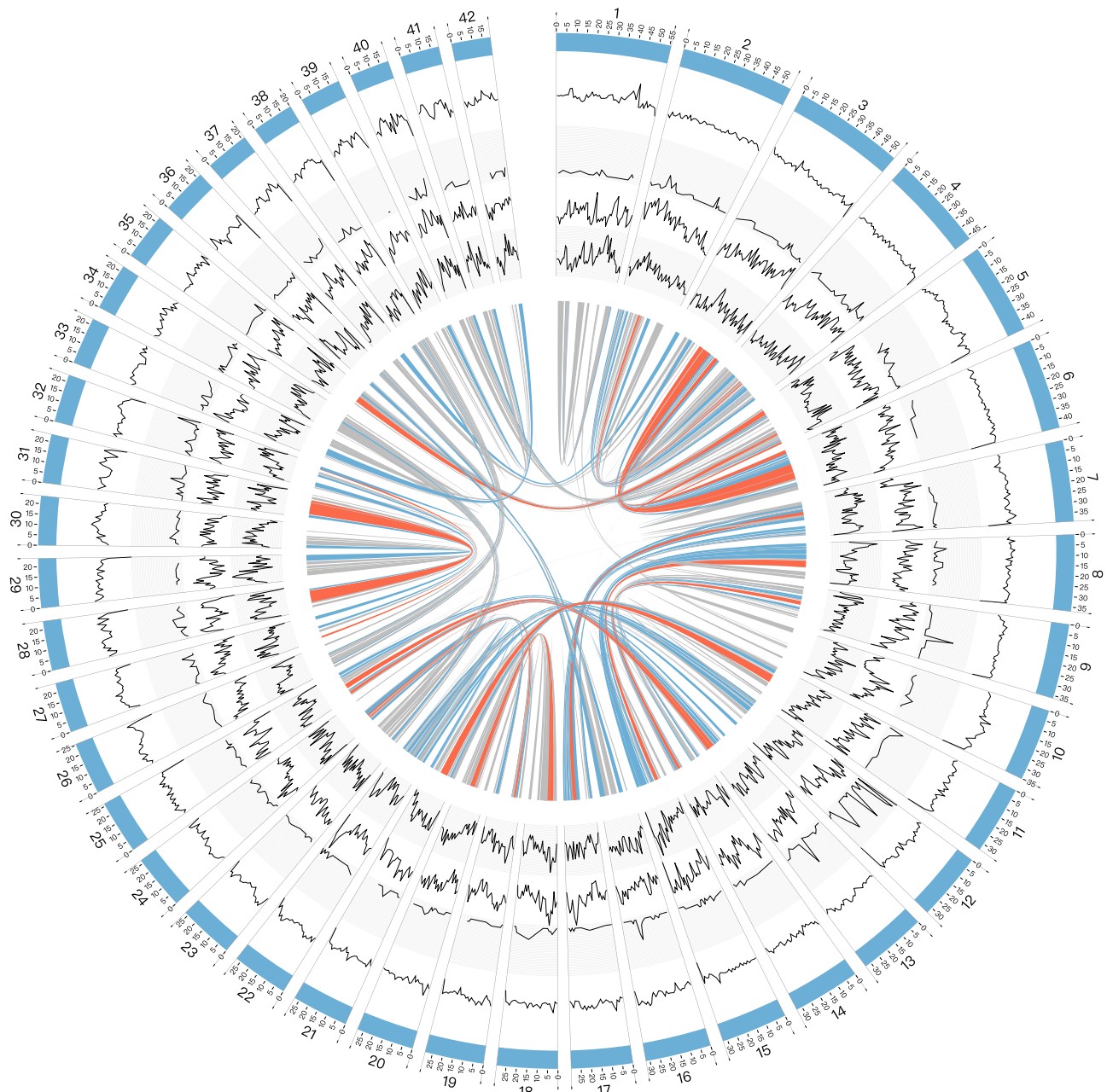

**Fig. 1 The genome overview of *Amynthas corticis*.** The outermost ring represents the longest 42 scaffolds. Then, from outside to inside, four rings represent the density distribution of different kinds of genomic elements annotated in the genome, including repeat sequences, non-coding genes, coding genes, and SNPs. Connecting lines in the center represent synteny blocks or duplicated genomic regions detected by MCScanX. The gray, blue, and red colors of the connecting lines represent alignment scores produced by MCScanX that are lower than 500, between 500 and 1000, and greater than 1000, respectively.

families for *A. corticis*, several other invertebrates (including annelids, flatworms, roundworms, molluscs, and insects) and vertebrates based on the TreeFam[38–40] database.

Through the analysis of CAFE[41,42], we identified 738 multiple-copy gene families with accelerated rates of gene duplication and loss in total. For these 738 multiple-copy gene families, the Spearman's rank correlation coefficient matrix of gene member count of species showed that *A. corticis* was dissimilar with its phylogenetic neighbors of annelids, but similar with flatworms, roundworms, and two snakes in vertebrates (Fig. 3a, Supplementary data file 4), which suggested that gene members in families with accelerated rates of gene duplication and loss made the key difference on physiology between *A. corticis* and other annelids.

Detailed analysis showed that in the genome of *A. corticis*, gene members in most of families with accelerated rates of gene duplication and loss (554/738, 75.07%) were expanded (Supplementary Fig. 4), rather than contracted (Supplementary Fig. 5) or stable (Supplementary Fig. 6), implying fast evolution of paralogs in the genome of *A. corticis*. Furthermore, principal component analysis (PCA) for the gene member count of species in families with accelerated rates of gene duplication and loss showed that *A. corticis* was dissimilar with other two annelids along the first component, even farther than flatworms, roundworms, and planaria (Fig. 3b, Supplementary data file 5). To find out which function drove *A. corticis* dissimilar with other two annelids, we first selected five subsets of gene families

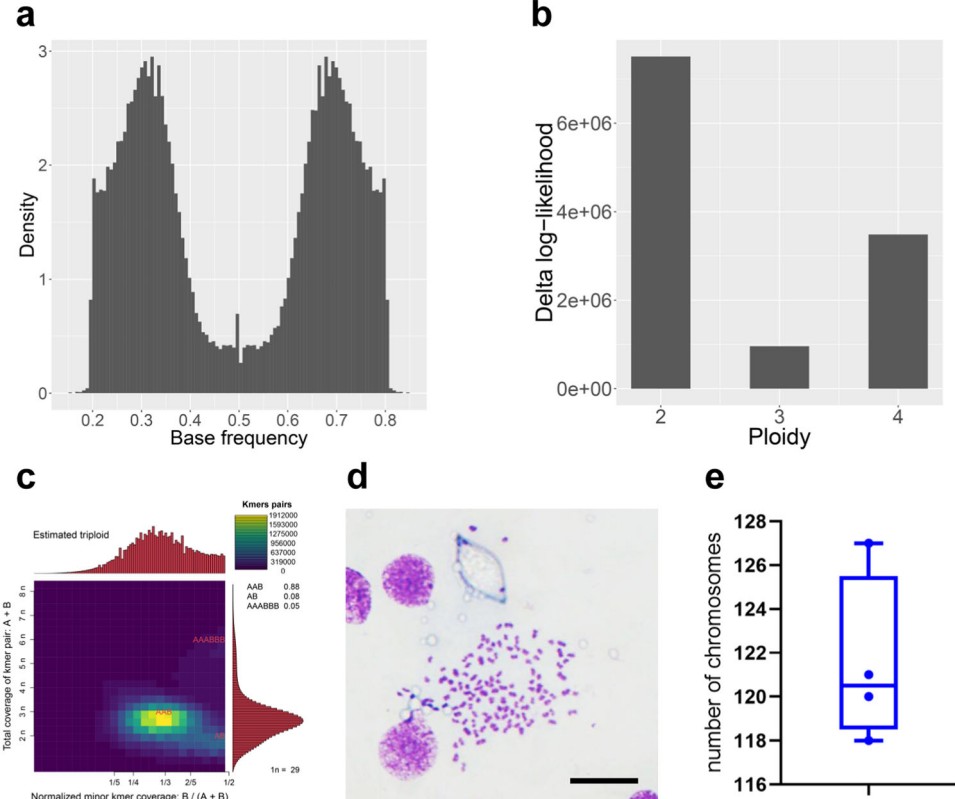

**Fig. 2 Proofs supporting that *Amynthas corticis* is triploid. a** The density distribution of allele frequency related to SNPs on bi-allelic loci, in which two peaks appear near the frequency of 1/3 and 2/3, respectively. **b** Delta log-likelihood values under hypotheses of diploid, triploid, and tetraploid, in which the smallest value under the hypothesis of triploid means the highest probability. **c** Heatmap for coverage pattern of heterozygous k-mer pairs, in which $X$ axis indicates the normalized minor k-mer coverage, while $Y$ axis indicates the total k-mer pairs coverage. The location on the heatmap colored by bright yellow represents that there are the most k-mer pairs (88%) supporting the normalized minor k-mer coverage of 1/3 and the total k-mer pairs coverage of 3n. **d** The image of chromosomes in one individual *A. corticis* produced in karyotype analysis. Scale bar, 10 μm. **e** The distribution of chromosome numbers in four *A. corticis* individuals determined via karyotype analysis.

whose variable loading in the first component is positive, including ones with the gene member count of *A. corticis* larger than that of other annelids and the mollusc (class_1_family), larger than that of other annelids, the mollusc and the flatworm (class_2_family), larger than that of other annelids, the mollusc, the flatworm, and the roundworm (class_3_family), larger than that of other annelids, the mollusc, the flatworm, the roundworm, and the snake (class_4_family), and larger than that of other annelids, the mollusc, the flatworm, the roundworm, the snake, and the planaria (class_5_family). Then, we conducted functional enrichment analysis[43,44] for each subset of gene families. There were the most functions enriched in class_5_-family (Fig. 3c, Supplementary data file 6). Referred to GOSlim[45], we found many functions related to immune system process (Supplementary Fig. 7), response to stress (Supplementary Fig. 8), homeostatic process (Supplementary Fig. 9) and several basic physiology functions, including anatomical structure development (Supplementary Fig. 10), signal transduction (Supplementary Fig. 11), and cell differentiation (Supplementary Fig. 12), were enriched in class_5_family. It indicated that *A. corticis* gained abundant gene content related to defensive process through the evolutionary path to live in various kinds of terrestrial habitats. Intriguingly, we found there were much more enriched functions related to immune system process (Supplementary Fig. 16), response to stress (Supplementary Fig. 17), and homeostatic process (Supplementary Fig. 18) in the genome *A. corticis* compared to those in the genome of *Eisenia fetida*[34] or *Eisenia andrei*[46] (Supplementary data file 7).

**Defensive responses to pathogenic *E. coli* O157:H7.** We incubated *A. corticis* in an artificial soil infected with pathogenic *E. coli* O157:H7, which has been used as a stressor in ecotoxicology tests of earthworms[47–51]. iTRAQ-based proteomics analysis and 16 S rDNA gene sequencing of *A. corticis* under *E. coli* O157:H7 exposure were performed on the third, seventh, and twenty-eighth day. This time course allowed us to profile the dynamic patterns of protein expression and gut microorganism constitution of the *A. corticis* in response to the pathogenic *E. coli* O157:H7. Following normalization of expression level and 16 S rDNA abundance, we conducted time series analysis[52,53] for both of data sets.

147 differentially expressed proteins were identified varied along the time course by iTRAQ method, and in which 39, 34, 33, and 41 proteins showed highest expression before incubation or on the third, seventh, and twenty-eighth day after incubation, respectively (Fig. 4a, Supplementary data file 8). This time-based expression pattern revealed subtle regulation of gene expression in *A. corticis* when facing this stressor. To further understand the biological sense underlining the time-based expression pattern, we annotated GOSlim terms and calculated their ratios for each set of genes with an expression peak at a specific time point. Through ranking ratios of the same GOSlim term among all sets of genes, we found discriminated functional bias at different time points (Fig. 4b, Supplementary data file 9). Before incubation, the ratio of GOSlim terms related to basic cell functions such as cytoskeleton organization, protein transport, transmembrane transport, and signal transduction, ranked highest, suggesting

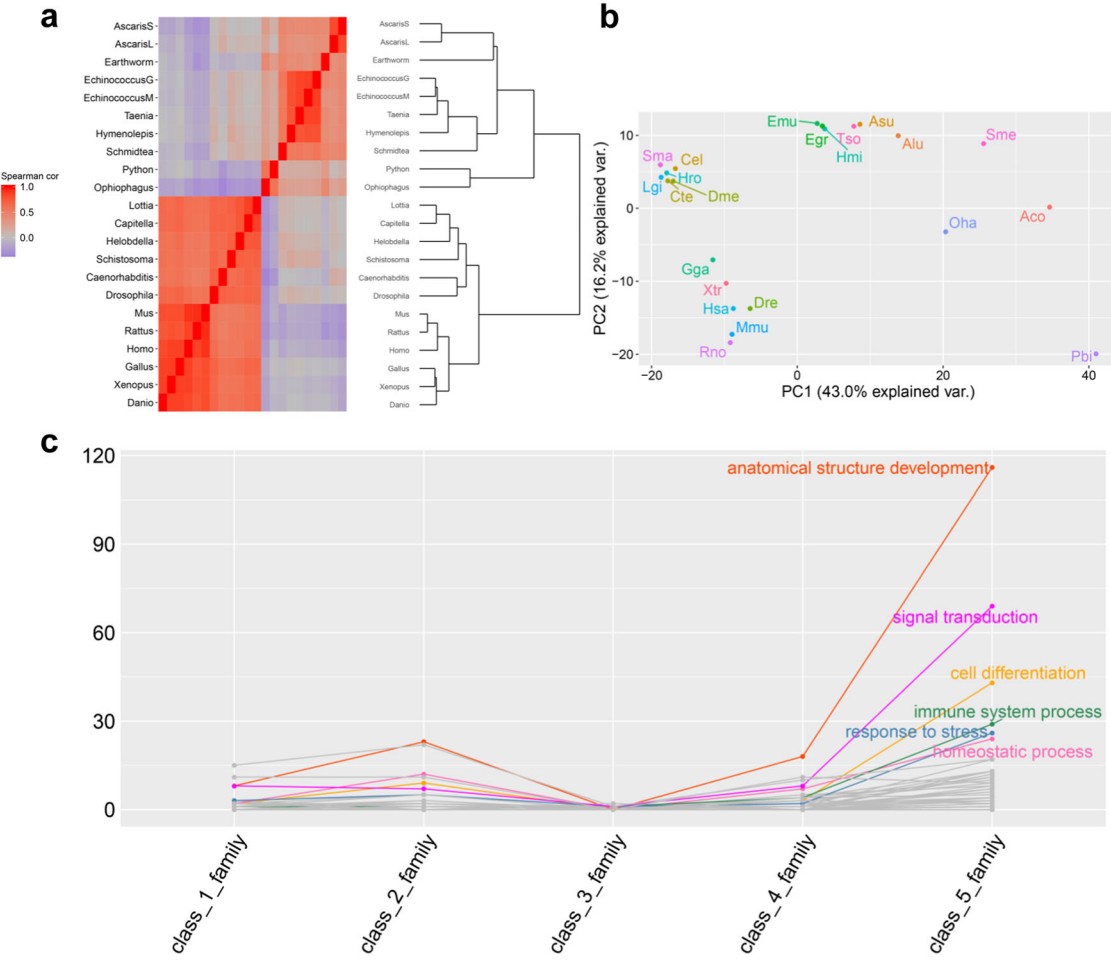

**Fig. 3 Characteristics of multiple-copy gene families with accelerated rates of gene duplication and loss. a** The heatmap and dendrogram plotted based on Spearman's rank correlation coefficient matrix of gene member count of species of multiple-copy gene families with accelerated rates of gene duplication and loss. In the heatmap, the redder the color is, the stronger the correlation is. To plot the dendrogram, the distance between each pair of species is defined as the difference value between 1 and the corresponding Spearman's rank correlation coefficient. **b** Plot of PCA for 22 species, in which the first two components account for 43% and 16% of variation, respectively. Components of species were calculated based on the count of gene member in families with significantly accelerated rates of gene duplication and loss. Along the first component, *Amynthas corticis* is dissimilar with other two annelids. **c** The count distribution of enriched functions in each subset of gene families with accelerated rates of gene duplication and loss, in which enriched functions under the same GOSlim term are plotted as one line. The GOSlim term is labeled if the number of enriched functions attributed to it is larger than 20 in class_5_family. In this figure, AscarisL or Alu *Ascaris lumbricoides*, AscarisS or Asu *Ascaris suum*, Caenorhabditis or Cel *Caenorhabditis elegans*, Capitella or Cte *Capitella teleta*, Danio or Dre *Danio rerio*, Drosophila or Dme *Drosophila melanogaster*, Earthworm or Aco *Amynthas corticis*, EchinococcusG or Egr *Echinococcus granulosus*, EchinococcusM or Emu *Echinococcus multilocularis*, Gallus or Gga *Gallus gallus*, Helobdella or Hro *Helobdella robusta*, Homo or Hsa *Homo sapiens*, Hymenolepis or Hmi *Hymenolepis microstoma*, Lottia or Lgi *Lottia gigantea*, Mus or Mmu *Mus musculus*, Ophiophagus or Oha *Ophiophagus hannah*, Python or Pbi *Python bivittatus*, Rattus or Rno *Rattus norvegicus*, Schistosoma or Sma *Schistosoma mansoni*, Schmidtea or Sme *Schmidtea mediterranea*, Taenia or Tso *Taenia solium*, Xenopus or Xtr *Xenopus tropicalis*.

*A. corticis* lived in a normal status. On the third day after incubation, the ratio of GOSlim terms related to defensive processes, including response to stress and immune system process, ranked highest. In addition to that, the ratio of GOSlim terms of cell population proliferation and cell adhesion also ranked highest, indicating cellular defensive process against pathogenic bacteria, such as expansion of phagocytes and assembly of encapsulation components[54,55]. On the seventh day after incubation, the ratio of GOSlim terms related to metabolic and catabolic process ranked highest such as carbohydrate metabolic process, small molecule metabolic process, cofactor metabolic process, cellular nitrogen compound metabolic process, and catabolic process, indicating activation of the humoral defensive process against pathogenic bacteria[54,55]. In particularly, the ratio of GOSlim term of symbiotic process also ranked highest, suggesting interactions between *A. corticis* and its gut

microorganisms strengthened at this time point. Finally, on the twenty-eighth day after incubation, the ratio of GOSlim terms of cell death, cell motility, cell differentiation, and homeostatic process firstly ranked, indicating the recovering status of *A. corticis* following innate immunity.

Abundance of 16 S rDNA of 28 microorganisms varied along the time course, and in which 16 S rDNA of 8, 5, and 15 microorganisms possessed the highest abundance before incubation, on the third and twenty-eighth day after incubation, respectively (Fig. 4c, Supplementary data file 10). There were the most microorganisms with high abundance on the twenty-eighth day after incubation, corresponding with the recovering status of *A. corticis* at this time point. Then, we identified functional categories and calculated their relative abundances for each set of microorganisms with the highest abundance at a specific time point[56,57]. Comparing the relative abundance of the

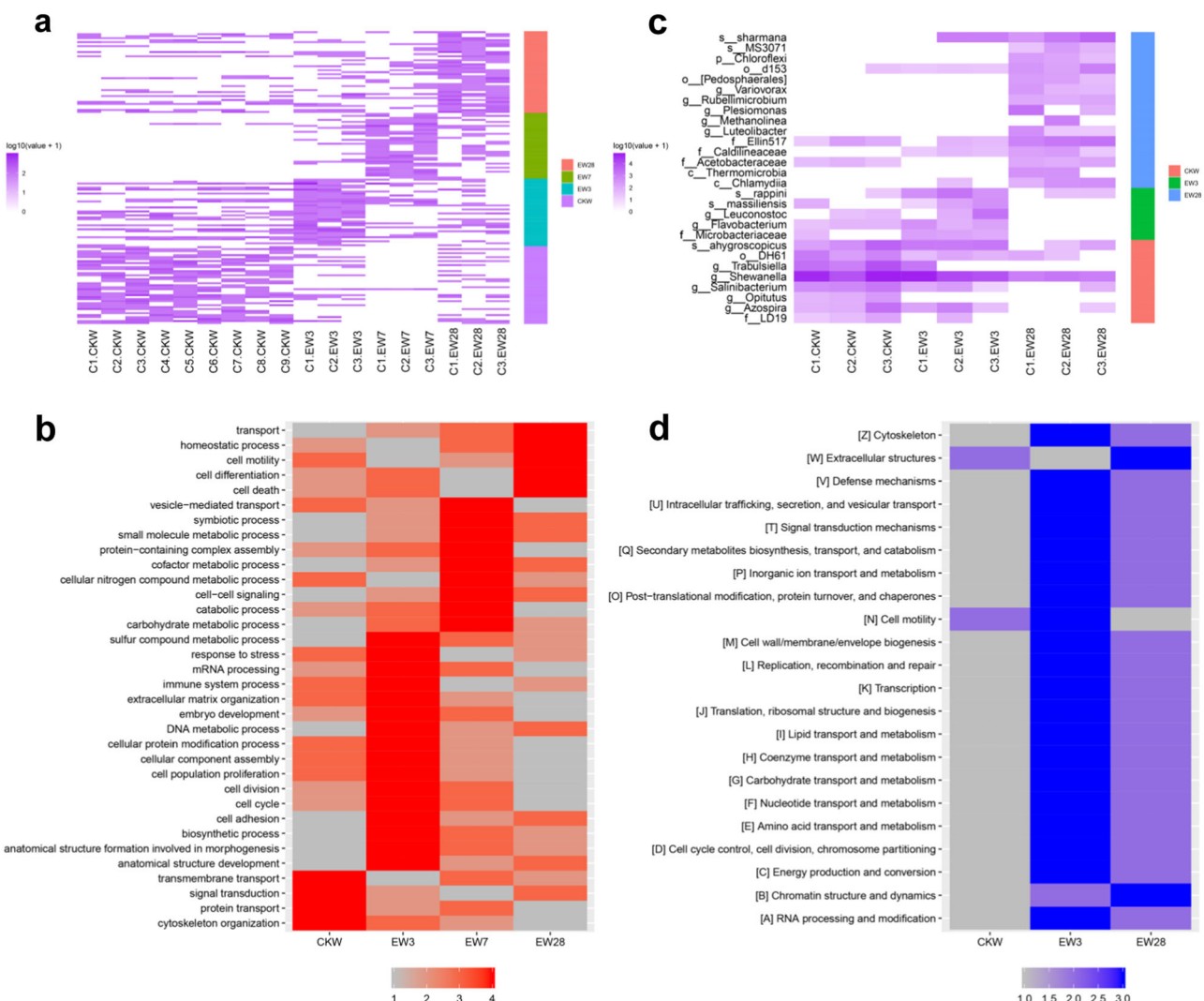

**Fig. 4 Defensive responses of *Amynthas corticis* to pathogenic *E. coli* O157:H7. a** Proteome of body tissues except the gut of *A. corticis* at different time points around the incubation in infected soil with pathogenic *E. coli* O157:H7. C(1–9).CKW are samples collected before incubation, while C(1–3).EW3, C(1–3).EW7, and C(1–3).EW28 are samples collected on the third, seventh, and twenty-eighth day after incubation, respectively. Gene set with expression peak before incubation (CKW), on the third (EW3), seventh (EW7), and twenty-eighth (EW28) day after incubation was aligned with the color track of purple, cyan, green, and red, respectively. **b** Ranked ratios of GOSlim terms among four sets of genes with expression peak before incubation (CKW), on the third (EW3), seventh (EW7), and twenty-eighth (EW28) day after incubation, in which color blocks of red represent ranks from 1 to 4. GOSlim terms were ordered based on the time point when the highest ratio appeared. **c** Gut microbiome of *A. corticis* at different time points around the incubation. C(1–3).CKW are samples collected before incubation, while C(1–3).EW3 and C(1–3).EW28 are samples collected on the third and twenty-eighth day after incubation, respectively. The microorganism set with the highest abundance before incubation (CKW), on the third (EW3) and twenty-eighth (EW28) day after incubation was aligned with the color track of red, green, and blue, respectively. **d** Ranked relative abundances of functional categories among three sets of microorganisms with the highest abundance before incubation (CKW), on the third (EW3) and twenty-eighth (EW28) day after incubation, in which color blocks of blue represent ranks from 1 to 3. Functional categories were ordered based on the time point when the highest relative abundance appeared.

same functional category between each set of microorganisms showed that 90% (20/22) functional categories ranked highest on the third day after incubation (Fig. 4d, Supplementary data file 11), indicating greatly active status of gut microorganisms with various kinds of functions except for extracellular structures (W) as well as chromatin structure and dynamics (B) induced by pathogenic bacteria at this time point.

We detected interactions between *A. corticis* and its gut microorganisms through weighted correlation network analysis (WGCNA)[58] based on the proteome of body tissues except the gut and the gut microbiome. Four interaction motifs were generated in total (Supplementary data file 12,13). In the first motif, 25 out of 26 elements possessed the highest expression or

abundance before incubation, indicating this motif played a key role in the normal living status (Fig. 5a). In this motif, the gene annotated with GOSlim of response to stress appeared as a hub (evm.model.Contig37.43, which is orthologous with the human gene of NRBF2). Microorganisms in this motif had two kinds of essential functions, including metabolism of lignocellulose related carbohydrates[59,60] and production of antibiotics[61–63]. In the second motif, 11 out of 17 elements possessed the highest expression or abundance on the third day after incubation, suggesting this motif was stimulated by the stress of pathogenic *E. coli* O157:H7 (Fig. 5b). In this motif, there were a gene functioning in immune system process (evm.model.Contig35.246, TOLLIP) and a microorganism attributed to the genus of

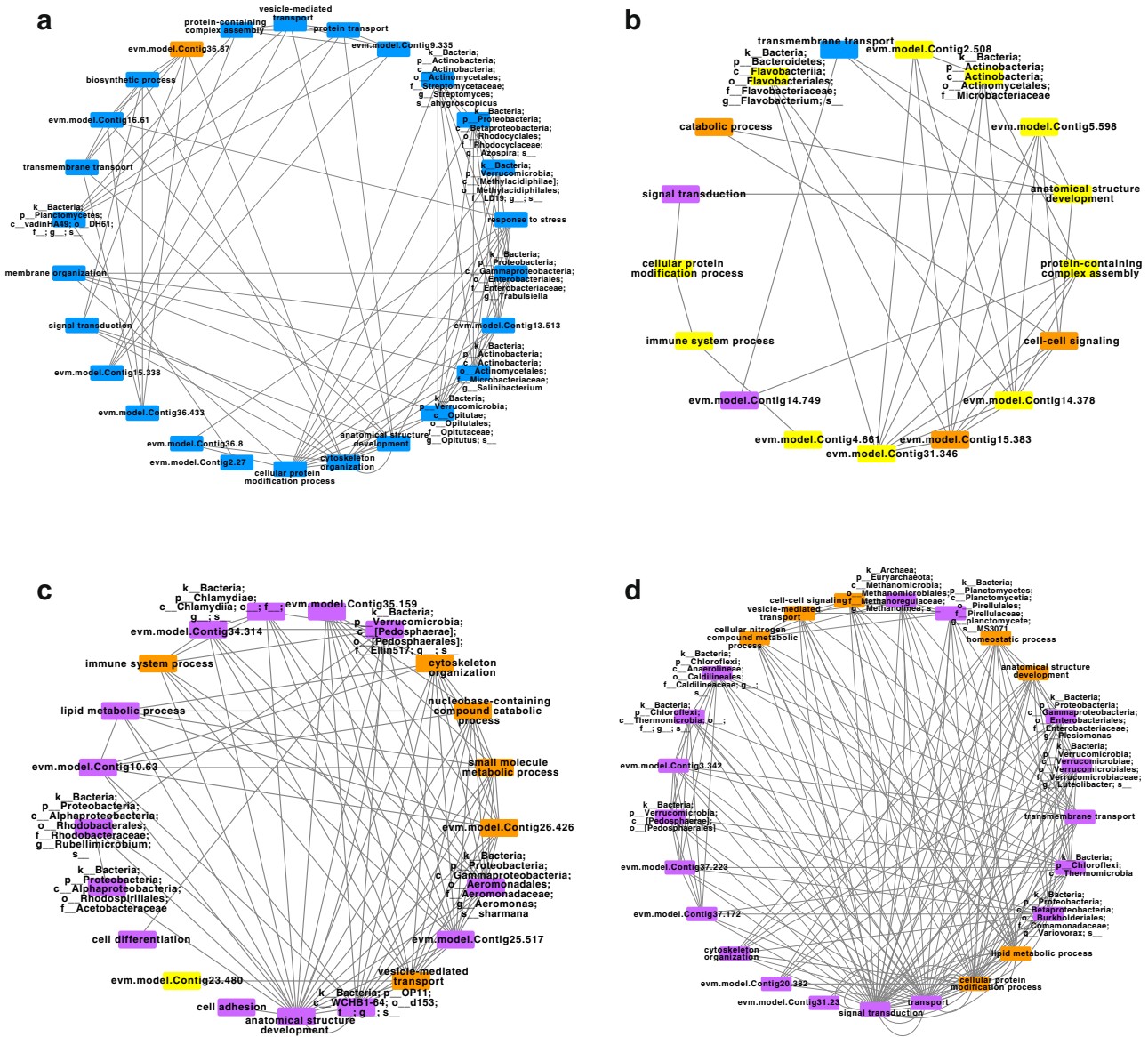

**Fig. 5 Interaction motif between *Amynthas corticis* genome and its gut microbiome.** In the first motif (**a**), the second motif (**b**), the third motif (**c**), and the fourth motif (**d**), the ID or GOSlim of the protein element and taxon of the microorganism element were labeled on each node, while the interaction between each two nodes was represented by a gray line connecting them. The node was colored with blue, yellow, orange, or purple if the corresponding element possessed the highest expression or abundance before incubation, on the third, seventh, or twenty-eighth day after incubation.

*Flavobacterium*, which distributes in the rhizospheres of agricultural crops and is associated with stimulation of plant resistance to disease[64–66]. In the third motif, 6 and 14 out of 21 elements possessed the highest expression or abundance on the seventh and twenty-eighth day after incubation, respectively (Fig. 5c). In this motif, most of genes were associated with defensive processes, such as immune system process (evm.model.Contig21.457, LNPEP/ENPEP/TRHDE/ANPEP/ERAP1/NPEPPS/ERAP2), small molecule metabolic process (evm.model.Contig20.337, ACSM1/ACSM4/C10orf129/ACSM5/ACSM3), nucleobase-containing compound catabolic process (evm.model.Contig13.182, SUPV3L1), lipid metabolic process (evm.model.Contig15.206, UGT3A2/UGT2B10/UGT2A1/UGT3A1/UGT2B15/UGT8/UGT2B4/UGT2B11/UGT2B7/UGT2B17/UGT2A3/UGT2B28), and cell adhesion (evm.model.Contig38.17, CHL1/L1CAM/NFASC/NRCAM). In addition, microorganisms involved in this motif were also associated with defensive processes, including production of bacteriocin[67,68], production of peptidase associated with activation and transport of antibiotic

bacteriocin or metalloendopeptidase[67,68], production of active oxidase and catalase[69,70], as well as modulating of immunity in insects[71,72]. In the fourth motif, 7 and 18 out of 25 elements had the highest expression level or abundance on the seventh and twenty-eighth day after *E. coli* O157:H7 exposure, respectively (Fig. 5d). In this motif, a gene functioning in homeostatic process (evm.model.Contig0.20, ZG16/ZG16B) and several microorganisms being plant growth-promoting rhizobacteria (PGPR) with the function of phytoremediation[73–78] interacted with each other, indicating the recovering status of *A. corticis* following innate immunity.

**Profile of defensive genes in the genome of *Amynthas corticis*.** The adaptability of earthworms, including the ecological plasticity of some commonly distributed invasive species and defensive ability has long motivated biologists seeking explanations of the phenomenon. For this reason, various kinds of well-determined defensive genes have been cloned and studied in detail (Supplementary data file 14), such as lysins (fetidin[19], lysenin[20], and

coelomic cytolytic factor, i.e., CCF[21]), antimicrobial proteins (lumbricin I[22], LBP/BPI[23], lysozyme[24]), toll-like receptors functioning in innate immune response against pathogens (mccTLR[25] and sccTLR[26]), proteins involved in response to oxidative stress (SOD[27], CAT[27], and CRT[28]) or metal stress (PCS[20]), detoxification proteins (GST[27] and CYP450[29]), and heat shock protein (HSP70[30]). However, there was still no holistic overview of earthworm defensive systems. Here, we systematically profiled copies of well-determined defensive genes from the assembled genome. Predicted genes were considered as orthologous hits of well-determined defensive genes if there were reciprocally best hits between them in BLAST search, while the genomic locations (Supplementary Fig. 13) and expression (Supplementary Fig. 14, Supplementary data file 15) of these genes were plotted.

In total, there were 111 orthologous hits identified from the assembled genome for 15 well-determined defensive genes, indicating abundant gene content related to defensive process in the genome of *A. corticis*. Further study showed that orthologous hits of 9 out of 15 (60%) genes appeared in expanded gene families, while orthologous hits of 8 out of 15 (53.3%) genes were located in duplicated regions detected by MCScanX[79] (Fig. 1), indicating the expansion trend of defensive genes. What is more, genomic regions containing orthologous hits of defensive genes had significantly higher density of SNPs than other regions (Fig. 1, the median density of SNPs in genomic regions containing orthologous hits of defensive genes vs. others: 19,601 vs. 17,546, Wilcoxon rank-sum test, *p*-value = 0.01), suggesting a high diversity of defensive genes in the population of *A. corticis*.

## Discussion

Comprehensive profiling for the genome of *A. corticis* helps to better understand features associated with adaptability of this species. First of all, triploid was suggested by ploidy estimation analysis based on the assembled complete chromosome-level genome and resequencing data, and also indicated by karyotype analysis. Triploid means more genomic materials for adaptive evolution and more copies of stress/defensive genes copies. In addition to that, comprehensive expansion of gene families related to immune system process, response to stress, homeostatic process, and other basic physiology functions made the key difference between *A. corticis* and other annelids. Based on detecting the proteome of its body and diversity as well as abundance of its gut microorganisms, it is observed that defensive process in *A. corticis* proceeded along the time course when facing the stress of pathogenic *E. coli* O157:H7 (from cellular defensive, humoral defensive to homeostatic process), suggesting ordered subtle regulation of defensive process in *A. corticis*. Particularly, close interactions between the genome of *A. corticis* and its gut microbiome were also observed, especially for the existence of microorganisms with functions of producing antibiotics[61–63], stimulating resistance to disease[64–66], modulating immunity[71,72], and assisting remediation[73–78], implying synergistic interactions between *A. corticis* and its gut microorganisms in the defensive process.

The assembly of a complete chromosome-level genome and systematic identification of gene copies in *A. corticis* can facilitate the usage of earthworms as model animals in ecotoxicology[80] and comparative immunology studies[55] to reveal the origins of sophisticated immunity in invertebrates, while providing abundant data for the earthworm research community. Beyond for their value in research communities, earthworms are also useful in monitoring environmental pollution[80], and furthermore, are considered as abundant sources of biological active compounds with potential usage in industrial and medical fields[55]. Obviously, the assembled complete chromosome-level genome and identified comprehensive gene set of *A. corticis* can promote the application of earthworm in these fields.

## Methods

**Genomic DNA isolation.** Adult earthworms were collected from the campus of China Agricultural University. Total DNA was extracted from one earthworm using the following phenol/chloroform protocol. A whole adult earthworm was homogenized to powder in liquid $N_2$. The powder was transferred to a 1.5-mL microcentrifuge tube and homogenized in 0.5 mL lysis buffer containing 200 mM Tris-HCl (pH 8.0), 20 mM disodium EDTA, 1% SDS, 2 mg/ml RNase A (Qiagen, Hilden, Germany), and 20 mg/ml Proteinase K (Merck, Kenilworth, NJ, USA). The homogenate was incubated at 56 °C for 1 h. After incubation, the homogenate was centrifuged at 4 °C and 12,000 rpm for 5 min. A total of 350 μL supernatant was transferred to a new 1.5-mL microfuge tube and 700 μL iso-amyl alcohol was added, mixed fairly vigorously, and centrifuged for 5 min at 4 °C and 12,000 rpm. Then, the supernatant was carefully removed and discarded, and 200 μl pH = 8.0 10 mM Tris-HCl was added to dissolve the DNA. Then, 200 μl phenol/chloroform/iso-amyl alcohol (25:24:1) was added, and the mixture was mixed fairly vigorously for 7 min and centrifuged at 4 °C and 12,000 rpm for 5 min; the upper phase was transferred to a new microfuge tube. An equal volume of chloroform/iso-amyl alcohol (24:1) was added, and the tube was mixed vigorously and then centrifuged at 4 °C and 12,000 rpm for 5 min; the upper phase was transferred to a new microfuge tube. Half of the sample volume of 7.5 mol/L ammonium acetate, and two volumes of cold 100% ethanol were added, mixed fairly vigorously, and allowed to precipitate for 2 min. The sample was centrifuged at 4 °C and 12,000 rpm for 10 min, and the supernatant was discarded after careful removal. The DNA precipitate was washed with 1 mL cold 70% ethanol, and centrifuged at 4 °C and 12,000 rpm for 5 min; the supernatant was carefully removed and discarded. The DNA pellet was air-dried at room temperature, dissolved in 50 μl 10 mM Tris-HCl, pH 8.0, and stored at −20 °C.

**Genome survey sequencing.** Illumina short-insert paired-end sequencing technology was used for genome survey sequencing. A DNA library was constructed according to the manufacturer's instructions (Illumina, San Diego, CA, USA). Sequencing was performed on the Illumina HiSeq 2000 platform. A total of 53 Gb raw data (101-bp paired-end reads) were generated, and 51 Gb clean data (40-fold coverage) remained after data quality control using fastp[81] (version 0.6) with the parameter "--length_required 100". The genome size was then estimated with Jellyfish[82] (version 2.2.6) using 21 as the k-mer size. In addition to the sequencing data used for the genome survey, two other short-insert libraries were sequenced on the Illumina HiSeq X Ten platform, generating 80 Gb of raw data (150-bp paired-end reads, 62-fold coverage), which were used for downstream analysis.

**PacBio SMRT sequencing.** For SMRT sequencing, genomic libraries with an insert size of about 20 kb were prepared using standard protocols (https://www.pacb.com/wp-content/uploads/2015/09/Procedure-Checklist-20-kb-Template-Preparation-Using-BluePippin-Size-Selection.pdf; Pacific Biosciences, Menlo Park, CA, USA). The prepared DNA libraries were assessed with an Agilent 2100 Bioanalyzer (Agilent Technologies, Santa Clara, CA, USA) and sequenced on the Pacific Biosciences Sequel platform at the Genome Center of Nextomics (Wuhan, China). A total of 68 Gb subreads (53-fold) were generated, with a mean length of 7471 bp.

**Hi-C library sequencing.** Three Hi-C libraries were prepared from whole blood cells as described previously. Briefly, the cells were fixed with formaldehyde and then the cross-linked DNA was digested with DpnI. Their sticky ends were biotin-labeled and proximity-ligated to form chimeric junctions, which were further processed into paired-end libraries and subjected to sequencing on the Illumina HiSeq X Ten platform. A total of 265 Gb of raw data (150 bp paired-end reads) were generated and used for Hi-C scaffolding.

**Genome assembly.** To achieve the best genome assembly quality, FALCON[83] (version 0.3) and Canu[84] (version 1.6, r8533) were used for contig assembly. The default parameters were used for both FALCON and Canu. The N50 values of the FALCON and Canu assemblies were 117,165 bp and 55,452 bp, respectively. Considering the greater N50 length, and similar assembly size to the estimated genome size, the FALCON assembly was used for further analysis. To obtain a more accurate assembly, contigs generated by FALCON were initially polished with the arrow algorithm from GenomicConsensus (version 2.3.2, https://github.com/PacificBiosciences/GenomicConsensus) using all PacBio subreads, and further processed with Pilon[85] (version 1.22) using all paired-end reads sequenced from the short-insert libraries described in section S2. Reads sequenced from Hi-C libraries were uniquely mapped to the polished contigs using the Burrows-Wheeler Aligner, BWA[86] (version 0.7) sample function with the default parameters. Based

on the mapping results, scaffolding was performed using Lachesis[87] (commit 2e27abb) software with the following parameters: "CLUSTER_N 43, ORDER_-MIN_N_RES_IN_SHREDS 15, CLUSTER_MIN_RE_SITES 50 ORDER_-MIN_N_RES_IN_TRUNK 100". Gaps in the scaffolded assembly were filled using PBJelly[88] (version 15.8.24) based on PacBio SMRT long reads. All scaffolds longer than 5 kb were used for downstream analysis.

## Completeness assessment for the assembled genome

*Mapping rates*. First, we applied the mapping rates of PacBio SMRT long reads described in section S3, and the Illumina short reads described in section S2, to the assembled genome as an indicator of the completeness of the assembled genome. Long reads were mapped using BLASR[89] (smrtlink version 5.0.1) with the default parameters, while short reads were mapped using BWA-MEM (version 0.7) with the default parameters.

*BUSCO*. Next, we used BUSCO[90] (version 3.0.2) with the parameters "-l metazoa_odb9 -sp fly" to assess the completeness of the assembled genome based on the Metazoan dataset. To get measures of "Average number of orthologs per core genes" and "% of detected core genes that have more than 1 ortholog", we run the same analysis via the web interface of gVolante[91].

## Heterozygosity estimation

*K-mer analysis*. Paired-end reads sequenced from the short-insert libraries described in section S2 were first filtered using fastq_quality_filter in FASTX-Toolkit (version 0.0.13, http://hannonlab.cshl.edu/fastx_toolkit/) with the parameters "-Q 33 -q 20 -p 70". The filtered reads were then used to perform k-mer analysis with GCE[92] (version 1.0), using 21 as the k-mer size and the parameters "-m 1 -D 8 -H 1 -b 1".

*SNP analysis*. We also estimated the heterozygosity of the assembled genome through SNP analysis. Paired-end reads sequenced from the short-insert libraries described in section S2 were mapped to the assembled genome using BWA-MEM (version 0.7) with the default parameters. SNPs were identified from mapping results using bcftools[93] (version 1.6) with the default parameters. The heterozygosity of the assembled genome was estimated as the average ratio of heterozygous SNPs identified in the various libraries.

## Ploidy estimation

Two methods were used to estimate the ploidy of earthworms based on sequence reads from the short-insert libraries discussed in section S2. First, earthworm ploidy was estimated using a statistical framework based on the density distribution of allele frequency related to SNPs on biallelic loci[31,32]. Additionally, the coverage of heterozygous k-mers of 21-bp length, generated with KMC software[33] (version 3.1.0), was analyzed and visualized using the smudgeplot package (version 0.1.3, https://github.com/KamilSJaron/smudgeplot) to estimate earthworm ploidy.

## Karyotype analysis

Sexually mature earthworms were collected from the campus of China Agricultural University and cultured in the laboratory. Colchicine solution of 0.04% concentration was injected into the body cavity 20–22 h before the next operation. Then, the testes were dissected, collected, and ground with forceps in 1.5-mL microcentrifuge tubes containing 0.8% physiological saline. The cell suspension was transferred to a new 1.5-mL microcentrifuge tube. Cells were collected through centrifugation at 1000 rpm for 6 min. The cells were washed with 1.5 mL physiological saline through centrifugation at 1000 rpm for 6 min. Cells were incubated in hypotonic solution (0.075 mol/L KCl) at 25 °C for 30 min. Cells were collected via centrifugation at 1000 rpm for 6 min. The supernatant was removed, and cells were prefixed with 1 mL fresh fixing solution (ethanol:glacial acetic acid, 3:1) at 4 °C, and centrifuged at 900 rpm for 5 min. Cells were fixed with 1 mL fresh fix solution at 4 °C for 1 h, then centrifuged at 900 rpm for 5 min. Cells were fixed for a second time with 1.5 mL fresh fixing solution at 4 °C for more than 12 h. Cells were fixed for the third time with 1.5 mL fresh fixing solution at 4 °C for 1 h. The cells were centrifuged at 900 rpm for 5 min and left in 0.2 mL of fixative. This cell suspension was dropped onto slides chilled on ice, and the slides were then dried by passing them through the flame of an alcohol lamp twice. Once dry, they were stained with Giemsa stain for 10 min and photographed under a microscope (final magnification, 160×).

## RNA isolation and sequencing

Total RNA of adult earthworms was extracted using TRIzol reagent[94], while mRNA was isolated using the kit PolyATtract mRNA Isolation System III (Promega, Madison, WI, USA). A total of 1.5 μg RNA per sample was used as the input material for preparation of individual RNA libraries. Libraries were constructed using the NEBNext® Ultra™ RNA Library Prep Kit for Illumina® (NEB, Beverly, MA, USA) following the manufacturer's recommendations. Polymerase chain reaction (PCR) products with insert sizes of 150–200 bp were purified using the AMPure XP system and library quality was assessed with the Agilent Bioanalyzer 2100 system. Finally, the library was clustered using the cBot Cluster Generation System and sequenced on the Illumina HiSeq 2000 sequencing platform to generate paired-end reads.

## Genome annotation

*Repeat masking*. Repeats in the assembled genome were annotated using RepeatMasker[95] (version 4.0.7) and RepeatModeler (version 1.0.11, http://www.repeatmasker.org). The assembled genome was used as input data to generate a new repeat library in RepeatModeler with the parameter "-engine ncbi". The newly generated repeat library and the original repeat library of "Metaphire sieboldi" stored in RepBase[96] (version 21.04) were then merged. Finally, the repeat sequences were masked using RepeatMasker with the default parameters and the merged repeat library.

*Non-coding RNA gene prediction*. The repeat-masked genome was scanned with Infernal[97] (version 1.1.2) using the default parameters, and CM models were collected from the Rfam database[98,99] to predict non-coding RNA genes. A total of 1179 regions were annotated as non-coding RNA genes, with a total length of 123,329 bp. The predicted non-coding RNA genes belong to 67 unique families.

*Protein-coding gene prediction*. Ab initio, homology-based and transcriptome-based approaches were combined to predict protein-coding genes in the assembled genome. First, Augustus[100] (version 2.5.5) was used for ab initio protein-coding gene prediction with the parameter "--species=fly". Then, homologous protein sequences of seven known Lophotrochozoa species (*Lingula unguis*, *Crassostrea gigas*, *Helobdella robusta*, *Octopus bimaculoides*, *Biomphalaria glabrata*, *Capitella teleta*, and *Lottia gigantea*) were retrieved from the UniProt database[101] and aligned to repeat-masked genome using Exonerate[102] (version 2.2.0) with the parameters "--model protein2genome --percent 80 -n 1", to predict protein-coding genes. Next, raw reads generated from RNA sequencing were filtered using fastp (version 0.6) with the parameter "--length_required 100", mapped to the assembled genome using Hisat2[103] (version 2.1.0) with the parameter "-k 1 --dta-cufflinks", and assembled with Cufflinks[104] (version 2.2.1) in genome guide mode with the parameter "--max-mle-iterations 1000". Finally, all protein-coding genes predicted using the above three approaches were combined using EVM[105] (version 1.1.1) with the parameters "--segmentSize 100000 --overlapSize 10000".

*Functional annotation of protein-coding genes*. Four approaches were used to conduct functional annotation for protein-coding genes. First, protein sequences of protein-coding genes were searched against SwissProt database[106] (release 2018_11) using BLASTP[107] (version 2.6.0 + ), and matches with E-values ≤ 1e-5 were retained. Then, the protein-coding genes were scanned with InterProScan using the default parameters with annotations obtained from the InterPro database[108] (version 5.32–71.0). Next, protein-coding genes were annotated by KAAS[109] with the default parameters and annotations obtained from the Kyoto Encyclopedia of Genes and Genomes (KEGG) database[110]. Finally, protein-coding genes were annotated with eggNOG-mapper[111] (version 1.0.3) using the default parameters with annotation based on all organisms collected in the eggNOG database (version 4.5.1).

## Construction of phylogenetic tree for species

*Single-copy gene families*. To construct the phylogenetic tree of species, single-copy gene families were generated. First, protein sequences of coding genes in 22 species, including *Amynthas corticis*, *Helobdella robusta*, *Capitella teleta*, *Echinococcus granulosus*, *Echinococcus multilocularis*, *Hymenolepis microstoma*, *Taenia solium*, *Schistosoma mansoni*, *Schmidtea mediterranea*, *Ascaris lumbricoides*, *Ascaris suum*, *Caenorhabditis elegans*, *Lottia gigantea*, *Drosophila melanogaster*, *Danio rerio*, *Xenopus tropicalis*, *Ophiophagus hannah*, *Python bivittatus*, *Gallus gallus*, *Mus musculus*, *Rattus norvegicus*, and *Homo sapiens*, were retrieved from several resources (Supplementary data file 16)[38–40,112–114] and searched against HMM models of gene families collected and built in TreeFam[38–40] (version 9.0) using hmmscan in the toolkit HMMER[115] (version 3.1b2) with the parameters "--noali --notextw -E 0.05 --domE 0.05 --incE 0.01 --incdomE 0.01 --cpu 4". A protein-coding gene was attributed to a gene family in TreeFam if it had a match with the corresponding HMM model with the minimum E-value below 0.01. Second, single-copy gene families were selected from attributed ones if the count of gene members in all species is 1. Given that gene loss and incomplete gene annotation of genomes, gene families with the absence of gene in up to 50% of all species except for *Amynthas corticis* were also selected as single-copy ones. In total, 114 single-copy gene families were selected. In addition to that, 11,485 gene families with multiple gene members in any of the 22 species were considered as multiple-copy ones and retained for further analyses. The annotation of human genes recorded in HGNC (HUGO Gene Nomenclature Committee) was used as functional description for gene families.

*Supermatrix*. Protein sequences of gene members in each selected single-copy gene family were extracted from the proteome of each species via SAMtools faidx[116]. Then, multiple sequences alignment was implemented via MAFFT[117] with the parameters "--retree 2 --maxiterate 1000". Next, ID of aligned protein sequences was replaced with the corresponding species name for each alignment. Finally, all alignments were concatenated and filled with gaps to construct a supermatrix via FASconCAT-G[118] (version 1.04) with default parameters.

*Phylogenetic tree of species*. The phylogenetic tree of species was built via raxmlHPC-PTHREADS-AVX in the standard toolkit of RAxML[119] (version 8.2.12) based on the supermatrix constructed in the previous step. The evolutionary model of PROTCATBLOSUM62 and bootstrap convergence criterion were employed. The phylogenetic tree of species with the best maximum likelihood score was retained and labeled with bootstrap support values. Divergence time along the phylogenetic tree of species was estimated using r8s[120] (version 1.81) with the penalized likelihood (PL) method and truncated Newton (TN) algorithm. Divergence times at key nodes of species were determined from records collected in TimeTree[121].

**CAFE analysis**. The amount of gene members in each species was counted for 11,485 multiple-copy gene families. Multiple-copy gene families in which all species have fewer than 100 gene members were selected for CAFE[122] (version 4.2) analysis to avoid non-informative estimation of parameters. Following that, the gene member count distribution of each multiple-copy gene family and the phylogenetic tree of species indicated with divergence time were taken as the input of CAFE to estimate global birth and death rates and infer the most likely gene member count at internal ancestry nodes with default parameters. Gene families with the family-wise $P$-value smaller than 0.05 were considered as ones with accelerated rates of gene duplication and loss.

**Correlation of gene member count**. Gene member counts of different species in gene families with accelerated rates of gene duplication and loss were extracted for the Spearman correlation test between species using R (version 3.4.1). The distance between each pair of species was defined as the difference value between 1 and the corresponding Spearman's rank correlation coefficient. Spearman's rank correlation coefficient values were plotted on the heatmap and distances between different pairs of species were used to generate the dendrogram.

**Classification of gene families**. Gene families with accelerated rates of gene duplication and loss were classified as expanded, contracted, or stable ones for each species by comparing the gene member count observed in the species and inferred for the most recent common ancestor of all species included in this analysis. The numbers of expanded, contracted, and stable gene families in each species were counted and plotted.

**PCA for gene member count**. The gene member count of each species in 738 families with accelerated rates of gene duplication and loss was extracted for principal component analysis (PCA) using R (version 3.4.1). The first two components of all species included in this analysis were plotted. The variance accounted by each component was shown on the plot.

**GO enrichment analysis**. 11,485 multiple-copy gene families used in CAFE analysis were annotated with GO terms of human genes and considered as the background set for GO enrichment analysis. Five subsets of gene families with positive variable loading in the first component of PCA were selected, including ones with the gene member count of *Amynthas corticis* larger than that in *Lottia gigantean*, *Helobdella robusta*, and *Capitella teleta* (class_1_family), larger than that in *Lottia gigantean*, *Helobdella robusta*, *Capitella teleta*, and *Echinococcus multilocularis* (class_2_family), larger than that in *Lottia gigantean*, *Helobdella robusta*, *Capitella teleta*, *Echinococcus multilocularis*, and *Ascaris lumbricoides* (class_3_family), larger than that in *Lottia gigantean*, *Helobdella robusta*, *Capitella teleta*, *Echinococcus multilocularis*, *Ascaris lumbricoides*, and *Ophiophagus hannah* (class_4_family), while larger than that in *Lottia gigantean*, *Helobdella robusta*, *Capitella teleta*, *Echinococcus multilocularis*, *Ascaris lumbricoides*, *Ophiophagus hannah*, and *Schmidtea mediterranea* (class_5_family). Gene families in each subset were annotated and considered as test sets. GO terms of human genes were retrieved from BioMark in Ensembl genome browser of release Ensembl Genes 95 and GRCh38[123]. GOSlim terms were retrieved from The Gene Ontology knowledgebase of release date 2019-07-06[43,44]. GO enrichment analysis was conducted using Fisher's exact test with adjustment for multiple comparisons performed in R (version 3.4.1). Only the result with a false discovery rate (FDR) smaller than 0.05 was considered for downstream analysis. Following that, each enriched GO term was mapped to its corresponding GOSlim term. Finally, the number of enriched GO terms attributed to each GOSlim term was counted and plotted for five subsets of gene families.

**Synteny identification**. The protein sequences of all protein-coding genes were aligned to themselves using BLASTP (version 2.6.0 +) with the parameters "-evalue 1e-10 -num_alignments 5". Then, synteny blocks along the whole genome were identified using MCScanX[79] with the default parameters based on BLASTP results.

**Protein mass spectrometry**

*Performance of mass spectrometry*. Protein samples were extracted from bodies of earthworms incubated with *E. coli* O157:H7, as discussed in section S19. Separation of protein samples was conduct through high-performance liquid chromatography

(HPLC; UltiMate 3000 RS NANO LC; Thermo Fisher Scientific, Waltham, MA, USA) with A solution of formic acid (0.1%) and B solution of formic acid acetonitrile (80%). The chromatographic column was balanced with 94% A solution. Separated protein samples were analyzed through mass spectrometry using a Q-Exactive HF mass spectrometer (Thermo Fisher Scientific).

*Analysis of protein mass spectrometry*. Protein mass spectrometry data generated using isobaric tags for relative and absolute quantitation (iTRAQ) was searched against the proteome using MSGFPlus[124] (version v2019.02.28) and processed to extract the counts of each protein with the Bioconductor package RforProteomics[125] (version 3.8). The count of each protein in each sample was divided by the sum of the counts of all proteins in the same sample and multiplied by $10^6$ for normalization. The normalized counts were considered to represent the expression of proteins. Time series analysis was conducted to identify proteins with varied expression along the time course via ImpulseDE[52,53], in which only results with the $p$-value smaller than 0.05 were considered. Identified proteins were clustered according to the time point when the peak of expression appeared, and proteins in each cluster were annotated with GO terms. GO terms were then mapped to GOSlim terms, while the ratio of GOSlim terms in each cluster of proteins were calculated and plotted.

**Sequencing and analysis of 16 S rDNA**

*Sequencing of 16 S rDNA*. Adult earthworms were collected from the campus of China Agricultural University. The collected earthworms were pre-incubated in a phytotron with in-situ soil and maintained under an air temperature of 25 °C and 60% substrate moisture. After 30 days, the earthworms were transferred to a PVC box ($217 \times 173 \times 98$ mm) containing infected substrate (in-situ soil infected with *Escherichia coli* O157:H7 at a concentration of $10^7$ CFU/g dry weight of soil), and incubated in the phytotron. Earthworms were dissected to obtain the gut contents. Then, 660 μL PB solution (112.87 mM $Na_2HPO_4$, 7.12 mM $NaH_2PO_4$) and 220 μL TNS solution (0.5 M Tris HCl at pH 8.0, 0.1 M NaCl, 10% SDS) were added to a 2-mL centrifuge tube with a 0.2 g sample of earthworm gut content. After 50 s vibration (6.5 m/s), the tube was centrifuged for 5 min at 14,000 rpm and 4 °C. Then, 750 μL liquid supernatant was transferred to a new centrifuge tube and extracted in 750 μL PCI solution (the ratio of phenol, chloroform, and isoamyl alcohol was 25:25:1) and precipitated in isopropanol. The V4 region of the bacterial 16 S rDNA gene was sequenced on the Illumina HiSeq 2500 sequencing platform to generate paired-end reads.

*Analysis of 16 S rDNA*. Pairs of reads sequenced from original DNA fragments were merged via FLASH[126] (version 1.2.7) to generate joined reads. Joined reads were filtered using the plugin deblur (https://github.com/qiime2/q2-deblur) in QIIME2[127] (version 2018.11) with the parameter "--p-trim-length 250" to obtain only those of high quality. Joined reads of high quality were clustered into operational taxonomic units (OTUs) via the plugin vsearch[128] in QIIME2 with the parameter "--p-perc-identity 0.99", producing representative sequences and a feature table. The feature table is a matrix that recodes the numbers of observed occurrences of different OTUs in various samples. Representative sequences were used to perform taxonomic classification through the feature-classifier[129] plugin in QIIME2, trained using reference sequences collected in the Greengenes database[130] (release 13_8). The number of observed occurrences of each OTU in each sample was normalized by dividing by the total number of observed occurrences of all OTUs in the sample and multiplying by $10^6$ as its sample specific abundance. Time series analysis was conducted to identify 16 S rDNA with varied abundances in the time course via ImpulseDE[52,53], in which only results with the $p$-value smaller than 0.05 were considered. Identified 16 S rDNA set was clustered according to the time point when the highest abundance appeared, and 16 S rDNA in each cluster was annotated with COGs (Clusters of Orthologous Groups)[57] through PICRUSt (Phylogenetic Investigation of Communities by Reconstruction of Unobserved States)[56].

**Weighted correlation network analysis**. Weighted correlation network analysis (WGCNA)[58] was conducted to detect interactions between the genome of *Amynthas corticis* and its gut microbiome. Firstly, the data of protein expression and 16 S rDNA abundance with variation along the time course were combined according to the time of sampling, centered by the mean and scaled by the standard deviation. Then, WGCNA was running to construct interaction network and separated the network to different motifs. Last, protein elements were labeled with ID or GOSlim, microorganism elements were labeled with taxon and both of them were colored according to the time point when the highest protein expression or 16 S rDNA abundance appeared.

**Orthologous hits of well-determined defensive genes**. 15 well-determined defensive genes were collected through paper review. Orthologous hits of these genes were identified from the genome of *Amynthas corticis* via the strategy of reciprocally best BLAST. If the predicted protein had the reciprocally best hit (i.e., the identity of which is larger than 30% and the $e$-value of which is smaller than 0.001) with a well determined defensive gene, it was considered as the orthologous hit of the defensive gene. Resulted orthologous hits were annotated with assigned

TreeFam families, the evolutionary status (whether expanded or not) of assigned TreeFam families, overlaps of synteny region and expression intensities. Difference of SNP densities between genomic regions located with orthologous hits of well-determined defensive genes and other genomic regions was tested through Wilcoxon rank-sum test.

*Statistics and reproducibility.* Differences between compared groups were evaluated by Wilcoxon rank-sum test, while functional enrichment analysis was conducted by Fisher's exact test using inhouse R scripts. Multiple testing correction was conducted by the method of Benjamini and Hochberg using inhouse R scripts. $P$-value $<0.05$ or FDR $< 0.05$ was considered to be statistically significant. Experiments were repeated at least 3 times to ensure reproducibility and promise the power for statistics analysis.

**Reporting summary**. Further information on research design is available in the Nature Research Reporting Summary linked to this article.

## Data availability

The assembled genome and annotation data reported in this study has been deposited in the Genome Warehouse (https://bigd.big.ac.cn/gwh) in National Genomics Data Center (NGDC)[131], Beijing Institute of Genomics (China National Center for Bioinformation), Chinese Academy of Sciences with accession number GWHAOSM00000000.1. The genome sequencing data, transcriptome sequencing data, and 16 S rDNA sequencing data have been deposited in the Genome Sequence Archive (https://bigd.big.ac.cn/gsa/) in NGDC with accession number CRA003278. The protein mass spectrum data has been deposited in the OMix (https://bigd.big.ac.cn/omix/) in NGDC with accession number OMIX226. The whole project has been registered in BioProject (https://bigd.big.ac.cn/bioproject/) in NGDC with accession number PRJCA002308.

## Code availability

We developed the code of pipeline used to assemble the genome of earthworm with version 1.0, which was uploaded to https://github.com/ewasm3/ew.

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

## Acknowledgements

We thank NextOmics (Wuhan, China) for generating the PacBio and Hi-C data. We also thank Yan Sun, Xin Zhou, Qingxiao Li, and Qiong Yu for comments on the manuscript. We also thank Yanxia Hu, Xichun Zhang, Guozhen Xu, Yanqin Liu, Chong Wang, Xiaoping Diao, Yuhong Gao, Shuaizhang Li, Yanrui Luo, Xuelian Liu, Lan Yao, Feifan Guo, and so on, who are all previous members of Sun's earthworm research laboratory. This work was supported by grants from the National Natural Science Foundation of China (No. 31172091 and No. 31801190), the Fundamental Research Funds for the Central Universities (No. 2018QC155 and 2018ZH003), and the program of Excellent Youth from Hebei Educational Department and Provincial Natural Fund (grant No BJ2017027 and C2019408050).

## Author contributions

X.W., Y.Zhao, and Z.J.S. designed and supervised research. X.W., Y.Zhang, Y.B.L., H.J., and Y.M.B. collected materials for sequencing and generated transcriptome data, proteomics, 16 S rDNA and karyotype data. M.M.K. performed the genome assembly, genome annotation, and heterozygosity estimation. Y.Zhao and Y.Y. performed the synteny analysis, species tree, transcriptome, proteomics, 16 S rDNA analysis. X.W. performed karyotype analysis. X.W., Y.Zhao, Y.Zhang, Y.F.Z., M.M.K., S.W.J., and Z.J.S. wrote the paper with assistance provided by co-authors.

## Competing interests

The authors declare no competing interests.
