## [Peer Review File · Communications Biology]

Reviewers' comments:

Reviewer #1 (Remarks to the Author):

This genome will be a very important resource for many scientific communities. The work is solid and the authors provide a clear and repeatable set of methods. The writing of the main paper is strong in some places (abstract and introduction), but has some major problems in others (e.g. many grammar issues in the results). I point out some of the issues, but do not touch on most. Below are issues that I raise with various sections of the paper and supplemental methods:

it's weird that there are no methods in the main paper? Is this the journal's format for genome reports or something? It seems like at least there should be at least some summary of methods used in the main paper (e.g. type of sequencing performed, assembly method, annotation) a-la Nature articles.

Supplement Line 215: "protein sequences of coding genes in 22 species ... were retrieved from several resources"

-- a supplemental table with URLs or accessions of starting data rather than resources would be appropriate to make these analyses repeatable.

supplement Line 396: "19 well determined defensive genes were collected through paper review"

-- a supplemental table with justification for identifying as defensive, DOIs of papers that were used to make this determination etc. would help solidify this analysis.

Haplotigs are a major problem with assemblies from long reads. There is no discussion of how haplotigs were controlled for. What were the BUSCO duplication scores for the final assembly (i.e. "Average # of orthologs per core genes" and "# of detected core genes that have more than 1 ortholog")? Are these indicative of haplotig presence?

Figure 1: The squiggly lines of the four inner rings representing the density distribution of repeats, noncoding genes, coding genes, and SNPs is not very helpful (is there any trends to be seen in the figure?). The figure is only referred to once in the text and that is in regards to number of coding genes, which the figure does not really provide any help to the sentence. The MCSCANX results in the middle are potentially interesting, but not really discussed in the paper.

Line120: "The constructed phylogenetic tree indicates that the evolutionary relationship of *A. corticis* and other species is consistent with the common sense."

-- "Common sense" is difficult to define. It would be more impactful to mention if the result is consistent with previous phylogenies (e.g. James and Davidson 2012 doi:10.1071/IS11012).

Line124: "away from" is unclear. It suggests a spatial analysis, which this is not.

Line126: "These two contradictory phenomena"

-- This is unclear, which "phenomena" are being referred to here. It would be more clear to point out that the result in relation to the phylogeny, suggest that "gene members in families with..."

Line127: "make the key difference between *A. corticis* and other annelids"

-- The "difference" being evaluated here is not clear.

Line 131-134: "which is consistent with its larger genome size."

-- This sentence is unclear. Isn't it expected that paralogs evolve fast? It's also not clear why this is consistent with larger genome size.

Line 137: "was away from other two annelids"

-- "away from" is unclear. It suggests a spatial analysis, which this is not
-- this should be fixed throughout the manuscript

Line 183: "firstly ranked"
grammar (fix throughout)

Line 199: "Abundance of 28 16S"
-- "Abundance of 28S and 16S"

Figure 3B. Taxa names on the tree are tiny. They should be larger

Line 255: Defensive genes
-- These need to be clearly defined and examples given

Line 256: "earthworm...due to the absence of a complete genome"
-- There is no mention of the 2 published earthworm genome papers (both on *Eisenia fetida*). Both of these showed many gene duplications suggesting that the expansion of the genome occurred prior to the split between . Neither address whether they are triploid or diploid. To better understand the evolution of triploidy in earthworms, It would be important to, and not so difficult to test if *Eisenia fetida* is triploid from these data.

Zwarycz AS, Nossa CW, Putnam NH, Ryan JF. Timing and scope of genomic expansion within Annelida: evidence from homeoboxes in the genome of the earthworm *Eisenia fetida*. *Genome biology and evolution*. 2016 Jan 1;8(1):271-81.

Paul S, Arumugaperumal A, Rathy R, Ponesakki V, Arunachalam P, Sivasubramaniam S. Data on genome annotation and analysis of earthworm *Eisenia fetida*. *Data in brief*. 2018 Oct 1;20:525-34.

Reviewer #2 (Remarks to the Author):

Comments

This paper is the first report of a whole genome for the pheretimoid earthworms which provide a reference genome and will be very helpful for the study of physiology, molecular phylogeny, molecular phylogeography and evolutionary biology of earthworm, I strongly recommend it being published in "Communication Biology" with some correction suggestions and comments as follow:

1. *Amyntas corticis* is a polymorphism species with a widely distribution in the world and the polymorphism is duo to its parthenogenesis and probably polyploidy. Generally, parthenogenesis results in different level of degeneration of male reproductive organs in earthworm, such as spermathecae (number, position, form, etc.), prostate gland (lacking, present or abbreviated) and sometimes male pore (present or lacking, in one side or two sides). There are almost all the cases of polymorphism in *Amyntas corticis* and the different level of morphs may be related to different level of polyploidy (Different level of polyploidy has been reported even in the same parthenogenetic species of earthworm- Shen et al., 2011). So, it is necessary to provide precise and complete

information on the material referring to this study, such as the collection site parameters, collection date, description of main morphological and anatomical characters, at least the characters concerned parthenogenesis, even at individual level.

2、 About the distribution of *Amyntas corticis*, there is not yet a reliable document, it is very probably origin from the south of China, its primarily distribution should be the south of China and the presence of populations in the north of China may be the recolonization after the last glaciation, or by physical introduction. It is difficult to say the distribution of this species in other parts of the world is only due to "human intervention", other factors may also affect this process, the term "physical dispersal" or "physical introduction" is more suitable.

3、 It will be interesting to compare the genome of *Amyntas corticis* with that of *Eisenia andrei*, which is a diploid amphigenesis species of Lumbricidae with also a widely distribution by physical introduction in the world and its genome has just be published (Shao et al., 2020). I wonder if the parthenogenesis and polyploidy could help the species of earthworm to enhance their adaptability for dispersal. Their widely distribution in the world may be simply due to physical dispersal. There are also some pheretimoid species with parthenogenetic reproduction which have a very restrictive geographical distribution.

4、 The differentially expressed proteins relate to many factors: the induction or inhibition of nuclear gene expression; change of one or more points in the course of transcription and translation; the toxicity to protein itself without relation to gene expression, etc. The results of this study reveal an integrated mechanism under the stress of *E.coli* O157, but the response of genes is occasional. For example, if we change the inoculation amount of *E.coli*, the differentially expressed protein category and amount will also change. Therefore, this case toxicological study is far from exploration for the extensive environmental adaption ability. The toxicogenomics care more concrete set of genes to different type of adverse factors. The more these gene amount and category in *Amyntas corticis* than that in other earthworm species maybe more valuable to explain its extensive distribution. In fact, adaptation acts on populations, it means that a species acquired some physiological or behavioral characters which are suitable for survival in a specific habitat by natural selection. Therefore, the results on an acute ecotoxicological test are just the emergency response of animal in the face of environment changes and it is far from demonstrating the adaptability of a species. So, I suggest to remove this part from the MS and the results of ecotoxicological test may be published separately.

Reviewer #3 (Remarks to the Author):

I think this manuscript, seems to represent the first well-annotated earthworm genome, will be quite valuable for all sorts of research. The manuscript also describes some interesting patterns that, with a bit more explanation, will be very interesting as well. It seems clear that *Amyntas* is doing something differently than other annelids whose genomes have been sequenced to date. Though I have some concerns about the authors' interpretations along these lines, I certainly think the manuscript represents important and interesting work.

I have included a marked-up copy of the manuscript that includes several minor comments and questions, and I have only a few substantive concerns. First, it was not clear to me from reading the manuscript whether or not the genomic data have been made publicly available. The manuscript does not discuss this, but prior to publication, information about availability of data must be included. I see from the Reporting Summary that the data are being made available via the NGDC, but the accession number returned no hits when I searched. Obviously, that will need to be changed prior to publication.

Second, I think it is very unfortunate that the whole earthworm was used for DNA extraction. These

are fairly large animals, and surely the entire worm was not necessary to extract enough DNA for genomic sequencing? Ideally, most of the animal would have been retained as a voucher specimen and placed in a museum or research collection. The authors do not discuss how they identified the earthworm as *A. corticis*. Fortunately, that should be fairly easy to do crudely after the fact using the COI barcode sequence. To that end, I would like to see a supplementary figure that shows the placement of the barcode sequence of the worm used in this study in a COI barcode phylogeny for *Amyntas*. Again, it is a shame that taxonomically important parts of the body of the worm were not retained and deposited in a museum collection – genome sequences must be tied to voucher specimens in research collections whenever possible.

Finally, as noted above, I question some aspects of the authors' interpretations of their findings, simply because I do not think we have sufficient comparative data yet to allow the authors to make their claims. To highlight this concern, I have extracted and slightly modified one of my main comments on the manuscript from the marked-up version – It makes sense that earthworms would be exposed to an array of microorganisms in the soil. However, it is not clear to me that pathogens are more common or diverse in soil than they are in, say, marine or freshwater sediments. It is also not clear to me that *Amyntas corticis* is unusual in its defensive ability in comparison to other earthworms. I certainly think it is interesting to study these defensive genes/proteins in *Amyntas*, but I think the authors should be more circumspect about what their findings might mean. It is possible that invasive earthworm species have more copies, and more variation, in their defensive genes than other earthworm species do, but it is also possible that ALL earthworms have more copies and more variation in their defensive genes than, say, leeches or *Capitella* or other spiralian animals have. Ideally, the authors would be able to compare their findings to other earthworm genomes. Unfortunately, though substantial work has been done on *Eisenia fetida*, that genome (<https://www.ncbi.nlm.nih.gov/genome/?term=eisenia>) does not seem to be annotated, though I think the authors should discuss the *Eisenia* genome at least briefly, to make clear that their genome sequence is not the first generated for an earthworm. In short, I think we are too early in the game to be able to attribute invasive ability to expansions or higher levels of variation in defensive gene families, and I think the authors should make that clear. I don't think that lessens the impact of their work very much, so it should be easy for them to do.

September 29, 2020

RE: Revision of manuscript COMMSBIO-20-1340-T

Wang et al.,

Genome reveals molecular mechanisms behind global distribution of *Amyntas corticis*

Dear reviewers,

We thank the Communications Biology for their careful comments. Those comments are valuable and helpful for revising and improving our manuscript, as well as the important guiding significance to our researches. We have studied comments carefully and have made correction which we hope meet with approval. Revised portion are marked in red in the revised manuscript. The responds to the reviewer's comments are as following:

Reviewers' comments:

Reviewer #1 (Remarks to the Author):

This genome will be a very important resource for many scientific communities. The work is solid and the authors provide a clear and repeatable set of methods. The writing of the main paper is strong in some places (abstract and introduction), but has some major problems in others (e.g. many grammar issues in the results). I point out some of the issues, but do not touch on most. Below are issues that I raise with various sections of the paper and supplemental methods:

it's weird that there are no methods in the main paper? Is this the journal's format for genome reports or something? It seems like at least there should be at least some summary of methods used in the main paper (e.g. type of sequencing performed, assembly method, annotation) a-la Nature articles.

Response: We appreciate the reviewer's suggestions and we have added some summary of methods to the main paper.

Supplement Line 215: "protein sequences of coding genes in 22 species ... were retrieved from several resources"

-- a supplemental table with URLs or accessions of starting data rather than resources would be appropriate to make these analyses repeatable.

Response: We appreciate the reviewer's helpful suggestion and provided a supplementary table listing URLs or accessions of protein sequences for 22 species (Supplementary Table 6).

supplement Line 396: "19 well determined defensive genes were collected through paper review"

-- a supplemental table with justification for identifying as defensive, DOIs of papers that were used to make this determination etc. would help solidify this analysis.

Response: We appreciate the reviewer's helpful suggestion and provided a supplementary table with justification for identifying as defensive and DOIs of related papers (Supplementary Table 5). Meanwhile, we removed 4 genes from the defensive gene set based on careful thought.

Haplotigs are a major problem with assemblies from long reads. There is no discussion of how haplotigs were controlled for. What were the BUSCO duplication scores for the final assembly (i.e. "Average # of orthologs per core genes" and "# of detected core genes that have more than 1 ortholog")? Are these indicative of haplotig presence?

Response: We appreciate the reviewer's helpful reminder. In this work, to solve the problem of haplotigs, we used next-generation sequencing data in high depth as well as third-generation sequencing data to polish the initially assembled genome. After polished, errors related to SNV/InDel and SV were corrected and primary contigs with higher coverage depth were retained. The step of genome polish was written in Methods of this manuscript. Then, we used BUSCO v3.0.2 to evaluate completeness of the polished genome. The program did not output information of "Average # of orthologs per core genes" and "# of detected core genes that have more than 1 ortholog". Instead, the program showed the number and fraction of various kinds of categories defined by BUSCOs, i.e. "C:91.2% [S:68.6%,D:22.6%], F:2.0%, M:6.8%, n:978", where "C" meant "complete BUSCOs", "S" meant "complete and single-copy BUSCOs", "D" meant "complete and duplicated BUSCOs", "F" meant "fragmented BUSCOs", "M" meant "missing BUSCOs" and "n" meant "total BUSCO groups searched". Herein, "D" should be similar with "# of detected core genes that have more than 1 ortholog" and an indicator associated with levels of gene duplication in the genome but not haplotig presence. The result associated with BUSCO was written in the manuscript.

Figure 1: The squiggly lines of the four inner rings representing the density distribution of repeats, noncoding genes, coding genes, and SNPs is not very helpful (is there any trends to be seen in the figure?). The figure is only referred to once in the text and that is in regards to number of coding genes, which the figure does not really provide any help to the sentence. The MCSCANX results in the middle are potentially interesting, but not really discussed in the paper.

Response: We appreciate the reviewer's helpful comments. Figure 1 was just used to give an overview of the assembled genome, while there were no special trends associated with distributions of repeats, noncoding genes, coding genes and SNPs displayed in this figure. However, we found genomic regions containing orthologous hits of defensive genes had significantly higher density of SNPs than other regions

(the median density of SNPs in genomic regions containing orthologous hits of defensive genes vs. others: 19,601 vs. 17,546, Wilcoxon rank-sum test, p-value = 0.01), which was written in the last part of Results with referring Figure 1. In relation to MCScanX results, we found orthologous hits of 9 out of 18 (50%) defensive genes were located in duplicated regions detected by MCScanX, indicating the expansion trend of defensive genes. This point was also written in the last part of Results with referring Figure 1. To make an appropriate referring of Figure 1 in Introduction, we added a sentence "Beside protein-coding genes, we also identified repeat sequences, noncoding genes and SNPs and displayed them along with 42 chromosome-level scaffolds (Fig. 1)" after the original sentence "To reveal the whole picture of molecular mechanisms behind the ecological plasticity and adaptation of *A. corticis*, we sequenced its genome and generated a complete 1.2 Gb assembly, including 42 chromosome-level scaffolds with N50 length of 31 Mb (Table 1 and Supplementary Fig. 3) and annotated a total of 29,256 protein-coding genes in the genome".

Line120: "The constructed phylogenetic tree indicates that the evolutionary relationship of *A. corticis* and other species is consistent with the common sense."-- "Common sense" is difficult to define. It would be more impactful to mention if the result is consistent with previous phylogenies (e.g. James and Davidson 2012 doi:10.1071/IS11012).

Response: We appreciate the reviewer's comment. To make a clear statement, we changed the original sentence "The constructed phylogenetic tree indicates that the evolutionary relationship of *A. corticis* and other species is consistent with the common sense" to "As expected, the constructed phylogenetic tree showed that *A. corticis* was within the lineage of annelids".

Line124: "away from" is unclear. It suggests a spatial analysis, which this is not.

Response: We appreciate the reviewer's reminder and changed the original phrase "away from" to "dissimilar with".

Line126: "These two contradictory phenomena"

-- This is unclear, which "phenomena" are being referred to here. It would be more clear to point out that the result in relation to the phylogeny, suggest that "gene members in in families with..."

Line127: "make the key difference between *A. corticis* and other annelids"

-- The "difference" being evaluated here is not clear.

Response: We appreciate the reviewer's helpful comments and changed the original sentence "These two contradictory phenomena imply that gene members in families with significantly accelerated rates of gene duplication and loss make the key difference between *A. corticis* and other annelids" to "When compared to the result in relation to the phylogeny, it suggested that gene members in families with significantly accelerated rates of gene duplication and loss made the key difference on physiology between *A. corticis* and other annelids".

Line 131-134: "which is consistent with its larger genome size."

-- This sentence is unclear. Isn't it expected that paralogs evolve fast? It's also not clear why this is consistent with larger genome size.

Response: We appreciate the reviewer's helpful comment and changed the original sentence "which is consistent with its larger genome size" to "implying fast evolution of paralogs in the genome of *A. corticis*".

Line 137: "was away from other two annelids"

-- "away from" is unclear. It suggests a spatial analysis, which this is not
-- this should be fixed throughout the manuscript

Response: We changed the original phrase "away from" to "dissimilar with" here, and then changed the original phrase "away from" to "dissimilar with" throughout the manuscript.

Line 183: "firstly ranked"

grammar (fix throughout)

Response: We changed the original phrase "firstly ranked" to "ranked highest" throughout the manuscript.

Line 199: "Abundance of 28 16S"

-- "Abundance of 28S and 16S"

Response: We are sorry it's unclear. The original phrase "Abundance of 28 16S" means abundance of 28 kinds of 16S rDNA genes. To make it clear we changed the original sentence "Abundance of 28 16S rDNA significantly varied along the time course, and in which 8, 5 and 15 16S rDNA possessed the highest abundance before incubation, on the third and twenty-eighth day after incubation, respectively" to "Abundance of 16S rDNA of 28 microorganisms significantly varied along the time course, and in which 16S rDNA of 8, 5 and 15 microorganisms possessed the highest abundance before incubation, on the third and twenty-eighth day after incubation, respectively".

Figure 3B. Taxa names on the tree are tiny. They should be larger

Response: We appreciate the reviewer's helpful suggestion and made taxa names on the tree larger in the new plot.

Line 255: Defensive genes

-- These need to be clearly defined and examples given

Response: We appreciate the reviewer's suggestion. We clarified definitions of and listed examples for different categories of defensive genes in the manuscript, while provided a supplementary table (Supplementary Table 5) recording detail information of each kind of defensive genes. We expanded the original sentence "For this reason, several well determined defensive genes have been cloned and studied in detail" as "For this reason, various kinds of well determined defensive genes have been cloned

and studied in detail (Supplementary Table 5), such as lysins (fetidin, lysenin and coelomic cytolytic factor, i.e. CCF), antimicrobial proteins (lumbricin I, LBP/BPI, lysozyme), Toll-like receptors functioning in innate immune response against pathogens (mccTLR and sccTLR), proteins involved in response to oxidative stress (SOD, CAT and CRT) or metal stress (PCS), detoxification proteins (GST and CYP450) and heat shock protein (HSP70)".

Line 256: "earthworm...due to the absence of a complete genome"

-- There is no mention of the 2 published earthworm genome papers (both on *Eisenia fetida*). Both of these showed many gene duplications suggesting that the expansion of the genome occurred prior to the split between . Neither address whether they are triploid or diploid. To better understand the evolution of triploidy in earthworms, It would be important to, and not so difficult to test if *Eisenia fetida* is triploid from these data.

Zwarycz AS, Nossa CW, Putnam NH, Ryan JF. Timing and scope of genomic expansion within Annelida: evidence from homeoboxes in the genome of the earthworm *Eisenia fetida*. *Genome biology and evolution*. 2016 Jan 1;8(1):271-81.

Paul S, Arumugaperumal A, Rathy R, Ponesakki V, Arunachalam P, Sivasubramaniam S. Data on genome annotation and analysis of earthworm *Eisenia fetida*. *Data in brief*. 2018 Oct 1;20:525-34.

Response: We appreciate the reviewer's helpful suggestion. We evaluated the ploid of earthworm *Eisenia fetida* based on its sequencing data provided by these papers and found that the earthworm *Eisenia fetida* is diploid. In addition, we deleted the original sentence "due to the absence of a complete genome" for more accurate representation.

Reviewer #2 (Remarks to the Author):

Comments

This paper is the first report of a whole genome for the pheretimoid earthworms which provide a reference genome and will be very helpful for the study of physiology, molecular phylogeny, molecular phylogeography and evolutionary biology of earthworm, I strongly recommend it being published in "Communication Biology" with some correction suggestions and comments as follow:

1、 *Amyntas corticis* is a polymorphism species with a widely distribution in the world and the polymorphism is duo to its parthenogenesis and probably polyploidy. Generally, parthenogenesis results in different level of degeneration of male reproductive organs in earthworm, such as spermathecae (number, position, form, etc.), prostate gland (lacking, present or abbreviated) and sometimes male pore (present or lacking, in one side or two sides). There are almost all the cases of polymorphism in *Amyntas corticis* and the different level of morphs may be related to different level of polyploidy (Different level of polyploidy has been reported even in the

same parthenogenetic species of earthworm- Shen et al., 2011). So, it is necessary to provide precise and complete information on the material referring to this study, such as the collection site parameters, collection date, description of main morphological and anatomical characters, at least the characters concerned parthenogenesis, even at individual level.

Response: We appreciate the reviewer's helpful suggestion and added information for the material referring to this study, including collection site, collection date, description of morphological characters and characters concerned parthenogenesis, with clear photos in this manuscript. The added paragraph is "We collected *A. corticis* from the campus of China Agricultural University in July 2017 (Supplementary Fig. 1a-b, Supplementary Fig. 2). *A. corticis* is a common species widely distributed in grassland of parks and schools. The length, width and body segment number of *A. corticis* are 124 mm, 3.5 mm and 105. One pair of male pores, which are apart from each other with one fourth of body circumference, and one single female pore locate in venter of the eighteenth and fourteenth body segment, respectively. Three small and circular genital papillae are present around each male pore (Supplementary Fig. 1c). Four pairs of spermathecae locate in venter of intersegments (5/6, 6/7, 7/8 and 8/9) with ovoid ampulla, straight stalk and blunt ovoid diverticulum (Supplementary Fig. 1d). Prostate gland of *A. corticis* is rudimentary to a small duct located in the eighteenth body segment, which is a phenotype concerned parthenogenesis".

2、 About the distribution of *Amyntas corticis*, there is not yet a reliable document, it is very probably origin from the south of China, its primarily distribution should be the south of China and the presence of populations in the north of China may be the recolonization after the last glaciation, or by physical introduction. It is difficult to say the distribution of this species in other parts of the world is only due to "human intervention", other factors may also affect this process, the term "physical dispersal" or "physical introduction" is more suitable.

Response: Following the reviewer's suggestion, we had changed the term "human intervention" to "physical dispersal".

3、 It will be interesting to compare the genome of *Amyntas corticis* with that of *Eisenia andrei*, which is a diploid amphigenesis species of Lumbricidae with also a wildly distribution by physical introduction in the world and its genome has just be published (Shao et al., 2020). I wonder if the parthenogenesis and polyploidy could help the species of earthworm to enhance their adaptability for dispersal. Their wildly distribution in the world may be simply due to physical dispersal. There are also some pheretimoid species with parthenogenetic reproduction which have a very restrictive geographical distribution.

Response: We appreciate the reviewer's suggestion and compared genomes of *Amyntas corticis*, *Eisenia andrei* and its close neighbor species *Eisenia fetida*. First, we evaluated ploidy for two earthworms in the genus of *Eisenia*. However, raw reads

sequenced for *Eisenia andrei* were not released on any public platform, thus we only evaluated ploidy for *Eisenia fetida*. We found *Eisenia fetida* is diploid though the same analysis method (Supplementary Fig. 15). Given that *Eisenia fetida* is close neighbor species of *Eisenia andrei*, it can be inferred that *Eisenia andrei* is also diploid with high probability. Then, we compared enriched functions of expanded gene families among these three earthworms and found counts of enriched functions associated with defenses, including response to stress, immune system process and homeostatic process, in *Amyntas corticis*, *Eisenia andrei* and *Eisenia fetida* were 131, 121, and 42, respectively. Also, we found enrichment levels of functions associated with response to stress (median odds ratio: 5.4 vs. 3.5, Wilcoxon rank sum test, p-value = 0.002, Supplementary Fig. 17) and immune system process (median odds ratio: 7.6 vs. 2.4, Wilcoxon rank sum test, p-value = $2e-12$, Supplementary Fig. 16) in *Amyntas corticis* were significantly higher than those in *Eisenia fetida*, meanwhile enrichment levels of functions associated with homeostatic process (median odds ratio: 8.4 vs. 6.6, Wilcoxon rank sum test, p-value = 0.029, Supplementary Fig. 18) in *Amyntas corticis* were significantly higher than those in *Eisenia andrei*. More intriguingly, we found enrichment levels of functions associated with response to stress (median odds ratio: 4.8 vs. 3.5, Wilcoxon rank sum test, p-value = 0.008, Supplementary Fig. 17) and immune system process (median odds ratio: 7.3 vs. 2.4, Wilcoxon rank sum test, p-value = $8.2e-12$, Supplementary Fig. 16) in *Eisenia andrei* were significantly higher than those in *Eisenia fetida*. Taken together, *Amyntas corticis* is not only triploid but also with more expanded gene families associated with defenses, implying its adaptability for dispersal was enhanced by both of these two characters. Beyond for that, it could be inferred that multiple genomic characters resulted from evolution influence adaptability for dispersal of earthworms together.

4、The differentially expressed proteins relate to many factors: the induction or inhibition of nuclear gene expression; change of one or more points in the course of transcription and translation; the toxicity to protein itself without relation to gene expression, etc. The results of this study reveal an integrated mechanism under the stress of E.coli O157, but the response of genes is occasional. For example, if we change the inoculation amount of E.coli, the differentially expressed protein category and amount will also change. Therefore, this case toxicological study is far from exploration for the extensive environmental adaption ability. The toxicogenomics care more concrete set of genes to different type of adverse factors. The more these gene amount and category in *Amyntas corticis* than that in other earthworm species maybe more valuable to explain its extensive distribution. In fact, adaptation acts on populations, it means that a species acquired some physiological or behavioral characters which are suitable for survival in a specific habitat by natural selection. Therefore, the results on an acute ecotoxicological test are just the emergency response of animal in the face of environment changes and it is far from demonstrating the adaptability of a species. So, I suggest to remove this part from the MS and the results of ecotoxicological test may be published separately.

Response: We appreciate the reviewer's helpful comment and suggestion. We included this part result associated with defensive responses to pathogenic *E. coli* O157:H7 in this manuscript to illustrate how the earthworm of *Amyntas corticis* regulated its gene expression and how gut microorganisms interacted with it when faced a stressor in detail. As is pointed by the reviewer, this result is occasional to some degree and far from exploration for the extensive environmental adaption ability. Thus, we should not and did not actually make the conclusion that this observed pattern was the mechanism behind environmental adaption ability of *Amyntas corticis*. However, we thought the value of these results lied in the attempt to uncover what happened in the earthworm when faced a stressor at the scale of whole genome as well as gut microbiome. In other words, this pattern can't be revealed much comprehensively without a well assembled genome. Therefore, we decide to retain this part result and appreciate the reviewer's suggestion again.

Reviewer #3 (Remarks to the Author):

I think this manuscript, seems to represent the first well-annotated earthworm genome, will be quite valuable for all sorts of research. The manuscript also describes some interesting patterns that, with a bit more explanation, will be very interesting as well. It seems clear that *Amyntas* is doing something differently than other annelids whose genomes have been sequenced to date. Though I have some concerns about the authors' interpretations along these lines, I certainly think the manuscript represents important and interesting work.

I have included a marked-up copy of the manuscript that includes several minor comments and questions, and I have only a few substantive concerns.

Response: We appreciate the reviewer's helpful suggestions and comments included in the marked-up copy of the manuscript. First, we corrected some typos and rewrote some texts followed the reviewer's suggestions. Second, we wrote point-to-point responses for all items below:

First, it was not clear to me from reading the manuscript whether or not the genomic data have been made publicly available. The manuscript does not discuss this, but prior to publication, information about availability of data must be included. I see from the Reporting Summary that the data are being made available via the NGDC, but the accession number returned no hits when I searched. Obviously, that will need to be changed prior to publication.

Response: We appreciate the reviewer's reminder. We have deposited the whole genome sequence data of *A. corticis* in the Genome Warehouse in NGDC¹⁵ under the accession number GWHAOSM00000000 which is publicly accessible at the website <https://bigd.big.ac.cn/gwh>. In addition, we also added a section named "Data availability" to describe that in the manuscript.

Second, I think it is very unfortunate that the whole earthworm was used for DNA extraction. These are fairly large animals, and surely the entire worm was not

necessary to extract enough DNA for genomic sequencing? Ideally, most of the animal would have been retained as a voucher specimen and placed in a museum or research collection. The authors do not discuss how they identified the earthworm as *A. corticis*. Fortunately, that should be fairly easy to do crudely after the fact using the COI barcode sequence. To that end, I would like to see a supplementary figure that shows the placement of the barcode sequence of the worm used in this study in a COI barcode phylogeny for *Amyntas*. Again, it is a shame that taxonomically important parts of the body of the worm were not retained and deposited in a museum collection – genome sequences must be tied to voucher specimens in research collections whenever possible.

Response: We appreciate the reviewer's helpful reminder. First, we added detail information for the worm used in this study, including collection site, collection date, description of morphological characters and characters concerned parthenogenesis, with clear photos in this manuscript. The added paragraph is "We collected *A. corticis* from the campus of China Agricultural University in July 2017 (Supplementary Fig. 1a-b, Supplementary Fig. 2). *A. corticis* is a common species widely distributed in grassland of parks and schools. The length, width and body segment number of *A. corticis* are 124 mm, 3.5 mm and 105. One pair of male pores, which are apart from each other with one fourth of body circumference, and one single female pore locate in venter of the eighteenth and fourteenth body segment, respectively. Three small and circular genital papillae are present around each male pore (Supplementary Fig. 1c). Four pairs of spermathecae locate in venter of intersegments (5/6, 6/7, 7/8 and 8/9) with ovoid ampulla, straight stalk and blunt ovoid diverticulum (Supplementary Fig 1d). Prostate gland of *A. corticis* is rudimentary to a small duct located in the eighteenth body segment, which is a phenotype concerned parthenogenesis". Then, we constructed a phylogenetic tree based on COI barcode sequences of the worm used in this study as well as other earthworms in *Amyntas* and provided it in a supplementary figure (Supplementary Fig. 2).

Finally, as noted above, I question some aspects of the authors' interpretations of their findings, simply because I do not think we have sufficient comparative data yet to allow the authors to make their claims. To highlight this concern, I have extracted and slightly modified one of my main comments on the manuscript from the marked-up version – It makes sense that earthworms would be exposed to an array of microorganisms in the soil. However, it is not clear to me that pathogens are more common or diverse in soil than they are in, say, marine or freshwater sediments. It is also not clear to me that *Amyntas corticis* is unusual in its defensive ability in comparison to other earthworms. I certainly think it is interesting to study these defensive genes/proteins in *Amyntas*, but I think the authors should be more circumspect about what their findings might mean. It is possible that invasive earthworm species have more copies, and more variation, in their defensive genes than other earthworm species do, but it is also possible that ALL earthworms have more copies and more variation in their defensive genes than, say, leeches or *Capitella* or other spiralian animals have. Ideally, the authors would be able to

compare their findings to other earthworm genomes. Unfortunately, though substantial work has been done on *Eisenia fetida*, that genome (<https://www.ncbi.nlm.nih.gov/genome/?term=eisenia>) does not seem to be annotated, though I think the authors should discuss the *Eisenia* genome at least briefly, to make clear that their genome sequence is not the first generated for an earthworm. In short, I think we are too early in the game to be able to attribute invasive ability to expansions or higher levels of variation in defensive gene families, and I think the authors should make that clear. I don't think that lessens the impact of their work very much, so it should be easy for them to do.

Response: We appreciate the reviewer's helpful suggestion and compared genomes of *Amyntas corticis*, *Eisenia andrei*¹³ and its close neighbor species *Eisenia fetida*^{1,14}. First, we evaluated ploidy for two earthworms in the genus of *Eisenia*. However, raw reads sequenced for *Eisenia andrei* were not released on any public platform, thus we only evaluated ploidy for *Eisenia fetida*. We found *Eisenia fetida* is diploid though the same analysis method. Given that *Eisenia fetida* is close neighbor species of *Eisenia andrei*, it can be inferred that *Eisenia andrei* is also diploid with high probability. Then, we compared enriched functions of expanded gene families among these three earthworms and found counts of enriched functions associated with defenses, including response to stress, immune system process and homeostatic process, in *Amyntas corticis*, *Eisenia andrei* and *Eisenia fetida* were 131, 121, and 42, respectively. Also, we found enrichment levels of functions associated with response to stress (median odds ratio: 5.4 vs. 3.5, Wilcoxon rank sum test, p-value = 0.002) and immune system process (median odds ratio: 7.6 vs. 2.4, Wilcoxon rank sum test, p-value = 2e-12) in *Amyntas corticis* were significantly higher than those in *Eisenia fetida*, meanwhile enrichment levels of functions associated with homeostatic process (median odds ratio: 8.4 vs. 6.6, Wilcoxon rank sum test, p-value = 0.029) in *Amyntas corticis* were significantly higher than those in *Eisenia andrei*. Taken together, *Amyntas corticis* is not only triploid but also with more expanded gene families associated with defenses compare to other earthworms in the genus of *Eisenia*, implying its adaption ability was enhanced by both of these two characters. Of course, as is pointed by the reviewer, it is too early in the game to be able to attribute invasive ability to expansions or higher levels of variation in defensive gene families. Thus, we weakened the opinion by changing the original sentence "Profiling of molecular mechanisms coded in the genome of *A. corticis* helps to better understand the basis of global distribution and stress defensive ability of this species" to "Profiling of molecular mechanisms coded in the genome of *A. corticis* helps to better understand features associated with adaption ability of this species".

Reviewer #3's comments and questions marked-up copy of the manuscript:

I would say that they are widely distributed around the world because they are an old group (originating in the Paleozoic) and they have been broadly distributed by continental drive. There are, however, several species of earthworm that have been introduced (and become invasive) in new habitats over the past few centuries. I think

that is what the authors are referring to here?

Response: Yes, this is what we meant and we expanded the original sentence “Earthworms (Annelida: Crassicitellata) are widely distributed around the world due to their great adaptability” to “Earthworms (Annelida: Crassicitellata) are widely distributed around the world due to their ancient origination as well as adaptation and invasion after introduced in new habitats over the past few centuries”.

I assume the authors will discuss how and why their new genome provides markedly more insight than and is superior to the *Eisenia fetida* genome? I imagine that long-read sequencing is superior to earlier approaches, but I think analysis of the *Eisenia* genome could also provide insights into invasion ability, as it has also been introduced accidentally and deliberately around the world.

Response: We assembled the genome of earthworm *Amyntas corticis* using the strategy combining third-generation long-read sequencing in high depth and Hi-C mapping, promising complete and accurate assembly. After assembling, we also polished the genome with high depth reads produced by next-generation sequencing and third-generation sequencing to remove errors associated with SNV/InDel and SV, respectively, further improving quality of the assembled genome. Therefore, scaffold N50 of the genome of *Amyntas corticis* (31 Mb) is much longer than that of *Eisenia fetida* (1,852 bp)¹. Then, we also evaluated ploidy and profiled enriched functions of expanded gene families for the genome of *Eisenia fetida* and found it was diploid and its expanded gene families enriched in fewer functions related to response to stress (16 vs. 41), immune system process (4 vs. 49) and homeostatic process (23 vs. 43) compared to the genome of *Amyntas corticis*, implying relative weak invasion ability of *Eisenia fetida*.

This would be interesting to examine in *E. fetida*, as it also has pungent defensive compounds.

Response: As suggested, we profiled enriched functions of expanded gene families for the genome of *E. fetida* and found its expanded gene families enriched in fewer functions related to response to stress (16 vs. 41), immune system process (4 vs. 49) and homeostatic process (23 vs. 43) compared to the genome of *Amyntas corticis*. Especially, for functions related to response to stress (median odds ratio: 3.5 vs. 5.4, Wilcoxon rank sum test, p-value = 0.002, Supplementary Fig. 17) as well as immune system process (median odds ratio: 2.4 vs. 7.6, Wilcoxon rank sum test, p-value = 2e-12, Supplementary Fig. 16), their enrichment levels in *E. fetida* were significantly lower than those in *Amyntas corticis*.

I do not understand this sentence.

Response: The original sentence in the manuscript is “Quantitative proteomic iTRAQ analysis shows 147 proteins and 16S rDNA sequences shows 28 microorganisms with significant responses to pathogenic *Escherichia coli* O157:H7”. To make it easier to understand, we rewrote it as “Quantitative proteomic iTRAQ analysis showed that expression of 147 proteins changed significantly in the body of *Amyntas corticis* as

well as 16S rDNA sequencing showed that abundance of 28 microorganisms changed significantly in the gut when the earthworm was incubated with pathogenic *Escherichia coli* O157:H7”.

I would write “clitellate” here, because oligochaetes are a paraphyletic group. They are not found in the deep sea, which covers most of the planet, so I have suggested an alternative wording.

Response: We appreciate the reviewer’s helpful suggestion and changed the word used here. The original sentence has been changed to “Earthworms are clitellate annelids distributed in virtually all terrestrial habitats except deserts and icecaps”.

Both of these concepts are reasonable, but they could also both be supported by theoretical and empirical citations.

Response: We appreciate the reviewer’s suggestion and added three citations to support these concepts in the manuscript. The original sentence was changed to “Polyploidy provides more genomic materials for evolving or expressing novel phenotypes^{2,3}, while parthenogenetic reproduction ensures the stability of those phenotypes⁴, which is beneficial when they are well-matched to a new environment”.

Greater than what? Greater than a typical earthworm, but I am not sure we know that. Many earthworms are invasive, and I suspect many others could be if transported around the world by humans. I think the invasive abilities of earthworms in general are worth exploring, but I do not think *Amyntas corticis* is special among earthworms in this regard.

Response: We appreciate the reviewer’s comment and changed the word “greater adaptation” to “adaptation”. In addition, we profiled and compared enriched functions of expanded gene families in several earthworms, while found counts of enriched functions associated with defenses, including response to stress, immune system process and homeostatic process, in *Amyntas corticis*, *Eisenia andrei* and *Eisenia fetida* were 131, 121, and 42, respectively. Also, we found enrichment levels of functions associated with response to stress (median odds ratio: 5.4 vs. 3.5, Wilcoxon rank sum test, p-value = 0.002) and immune system process (median odds ratio: 7.6 vs. 2.4, Wilcoxon rank sum test, p-value = 2e-12) in *Amyntas corticis* were significantly higher than those in *Eisenia fetida*, meanwhile enrichment levels of functions associated with homeostatic process in *Amyntas corticis* were significantly higher than those in *Eisenia andrei* (median odds ratio: 8.4 vs. 6.6, Wilcoxon rank sum test, p-value = 0.029), reflecting relative stronger invasive ability of *Amyntas corticis* than other two earthworms.

I have not heard of this before. I hope this is explained in more details.

Response: The original sentence “In addition to the genome, we also performed iTRAQ-based quantitative proteomics analysis to detect differentially expressed proteins of *A. corticis* and sequenced 16S rDNA from its intestinal tract after treatment with a pathogenic *Escherichia coli* stress O157:H7, which mimics severely adverse

environmental factors, to reveal how this earthworm genome functions when facing stresses” meant we adopted pathogenic *Escherichia coli* O157:H7 as a stressor in ecotoxicology tests of earthworms as did in previous studies⁵⁻⁹. To make it more clear, we changed the original sentence to “In addition to the genome, we also performed iTRAQ-based quantitative proteomics analysis to detect differentially expressed proteins of *A. corticis* and sequenced 16S rDNA from its intestinal tract after treatment with a pathogenic *Escherichia coli* stress O157:H7 to reveal how this earthworm genome functions when facing stresses”.

I am not very familiar with these calculations, but should this be “highest”?

Response: We double checked the result and this should be “lowest”. Here, we calculated delta log-likelihood between the free model estimated based on real data and fixed models of diploid, triploid and tetraploid, respectively. The calculated value is lower, the probability to be the specific ploid is larger². We found delta log-likelihood between the free model estimated based on our data and fixed model of triploid was lowest, thus we made the conclusion that *Amyntas corticis* is triploid.

How does the *Amyntas* genome compare to the *Eisenia* genome -- <https://www.ncbi.nlm.nih.gov/genome/?term=eisenia+fetida>?

Response: The size of *Eisenia fetida* is 1.05 Gb according to its best assembly¹. Thus, the genome of *Amyntas corticis* is also greater than that of *Eisenia fetida* by 1.14 fold.

The authors may want to explore two recent papers that explored gene family expansions in annelids and other spiralian -- <https://link.springer.com/article/10.1007%2Fs00239-020-09949-x> and (especially) <https://onlinelibrary.wiley.com/doi/abs/10.1111/jeb.13439> Some of these gene family expansions appear to be related to invasion of freshwater habitats by the ancestral clitellate annelid.

Response: We appreciate the reviewer's suggestion and read through these two papers. They provided strong supports to the point that expansion of gene families with specific function in a species was related to its invasion of new habitats. We expanded the original sentence “We constructed gene families for *A. corticis*, invertebrates, including annelids, flatworms, roundworms, mollusc and insect, and vertebrates based on the database of TreeFam” to “Previous studies have demonstrated that expansion of sodium-potassium pump alpha-subunit and several other gene families in leech happened at the same time of its transition from marine to freshwater habitats and might play important roles in diversification of annelid into freshwater habitats. To reveal how expansion of gene families affected earthworm *A. corticis*, we constructed gene families for *A. corticis*, invertebrates, including annelids, flatworms, roundworms, mollusc and insect, and vertebrates based on the database of TreeFam”.

Well, this is true, in that the annelids group together, but in other ways, that is a very

strange animal phylogeny.

Response: We checked the species tree displayed in the database of TreeFam (<http://www.treefam.org/browse#tabview=tab2>) and found most of partitions of our constructed tree are the same with it except for the root of flatworm lineage. However, the position of flatworm lineage is also not explicit in the species tree of TreeFam.

Usually this term (evolutionary rate) refers to substitution rate, not rate of gene duplication or loss. That should be clarified. Again, I would write something like “rate of gene duplication” or “rate of gene family growth” or something like that here.

Response: We appreciate the reviewer’s helpful reminder and changed the word “evolutionary rate” to “rates of gene duplication and loss”.

Environmental factors affect the evolutionary paths of all organisms, so this does not really say much.

Response: We appreciate the reviewer’s comment and removed this meaningless sentence “in which environmental factors might affect the evolutionary path of *A. corticis*”

See my comment above about this term.

Response: We appreciate the reviewer’s helpful reminder and changed the word “evolutionary rate ” to “rates of gene duplication and loss”.

The Horn et al. papers mentioned above are directly relevant to this discussion.

Response: We appreciate the reviewer’s helpful reminder and cited these two papers after the previous sentence “Previous studies have demonstrated that expansion of sodium-potassium pump alpha-subunit and several other gene families in leech happened at the same time of its transition from marine to freshwater habitats and might play important roles in diversification of annelid into freshwater habitats”.

This is awkwardly worded.

Response: We appreciate the reviewer’s comment and changed the original sentence “Through mapping GO terms to GOSlim, we found many functions attributed to GOSlim of immune system process (Supplementary Fig. 7), response to stress (Supplementary Fig. 8), homeostatic process (Supplementary Fig. 9) and several basic physiology functions, including anatomical structure development (Supplementary Fig. 10), signal transduction (Supplementary Fig. 11) and cell differentiation (Supplementary Fig. 12), were significantly enriched in class_5_family” to “Referred to GOSlim, we found many functions related to immune system process (Supplementary Fig. 7), response to stress (Supplementary Fig. 8), homeostatic process (Supplementary Fig. 9) and several basic physiology functions, including anatomical structure development (Supplementary Fig. 10), signal transduction (Supplementary Fig. 11) and cell differentiation (Supplementary Fig. 12), were significantly enriched in class_5_family”.

I do not see how this statement necessarily follows what proceeds it. The authors basically say that many different functions are represented in the class 5 family (i.e., genes that show significant expansions in the lineage leading to *Amyntas*). That is not surprising, and all of the ones they list - immune system, stress response and homeostasis - make sense for a genome from the only terrestrial annelid that they included. Surely some of these genes are related to the invasion of land. The authors mention a “defensive process”, which I suppose applies here (especially to the immune response), but “molecular weapons? seems out of place? Also, what is the hostile environment? The terrestrial habitat?

Response: We appreciate the reviewer’s helpful reminder and question. To make a more appropriate and clear statement here, we changed the original sentence “It indicated that *A. corticis* gained abundant gene content related to defensive process through the evolutionary path and thus equipped itself with large number of molecule weapons to live in hostile environments” to “It indicated that *A. corticis* gained abundant gene content related to defensive process through the evolutionary path to live in various kinds of terrestrial habitats”.

It is really only one stressor - pathogenic *E. coli*.

Response: We appreciate the reviewer’s helpful reminder and changed the original sentence “facing to stresses” to “facing this stressor”.

This is unclear. I assumed that “firstly ranked” meant “ranked highest”, but how can multiple terms rank highest?

Response: We appreciate the reviewer’s question and changed the original phrase “firstly ranked” to “ranked highest”. The “rank” we mentioned here meant the sort of ratios of each GOSlim term among four time points (before incubation, three-day, seven-day and twenty eight-day after incubation) but not the sort of all GOSlim terms.

A different term should be used... See comments above.

Response: We changed the original phrase “ranked firstly” to “ranked highest”.

I would not write “appeared”... surely those interactions were there the whole time. I would write that the nature of those interactions seems to have changed somehow.

Response: We appreciate the reviewer’s helpful suggestion and changed the original word “appeared” to “strengthened”.

This is not clear. 28 variants of 16S? What does the 28 refer to? Again, what do these refer to?

Response: We appreciate the reviewer’s questions. To make it clear we changed the original sentence “Abundance of 28 16S rDNA significantly varied along the time course, and in which 8, 5 and 15 16S rDNA possessed the highest abundance before incubation, on the third and twenty-eighth day after incubation, respectively” to “Abundance of 16S rDNA of 28 microorganisms significantly varied along the time course, and in which 16S rDNA of 8, 5 and 15 microorganisms possessed the highest

abundance before incubation, on the third and twenty-eighth day after incubation, respectively". In this sentence, 28 and other numbers meant counts of microorganism species.

I do not know what this means. Can the authors clarify this statement?

Response: This statement of "There were the most 16S rDNA possessed the highest abundance on the twenty-eighth day after incubation" meant there were the most microorganisms with high abundance detected in the gut of earthworm on the twenty-eighth day after incubation. To make the statement clear, we changed the original sentence to "There were the most microorganisms with high abundance on the twenty-eighth day after incubation".

Here is this term again, but I am still not sure what it means.

Response: We changed the original term "firstly ranked" to "ranked highest".

Which gut microorganisms? Can they be summarized here?

Response: We found gut microorganisms with various kinds of functions (20/22 functional categories recorded in the database of Clusters of Orthologous Genes, COG^{3,4}) had high abundance on the third day after incubation. Thus, we changed the original sentence "indicating greatly active status of gut microorganisms induced by pathogenic bacteria at this time point" to "indicating greatly active status of gut microorganisms with various kinds of functions except for extracellular structures (W) as well as chromatin structure and dynamics (B) induced by pathogenic bacteria at this time point".

This is unclear. Should it be "the gut (removed from the body)" or "body tissues except the gut"? I assume this is clarified in supplementary information, but it should be clear here.

Response: We appreciate the reviewer's suggestion and changed the original phrase of "gut removed body" to "body tissues except the gut".

This is awkwardly worded and is unclear.

Response: We appreciate the reviewer's comment and changed the original phrase "a gene annotated with GOSlim of immune system process" to "a gene functioning in innate immune system process".

This seems oddly phrased to me.

Response: We changed the original phrase "environmental plasticity" to "ecological plasticity".

Please check the wording here.

Response: We changed the original phrase "the ability of stress defensive" to "defensive ability".

I would like the authors to describe clearly what they mean by “defensive genes”. Many of the citations refer to antimicrobial compounds. Many readers will not be familiar with these compounds, and I think the authors should discuss them in the introduction.

Response: We appreciate the reviewer’s suggestion. We clarified definitions of and listed examples for different categories of defensive genes in the manuscript, while provided a supplementary table (Supplementary Table 5) recording detail information of each kind of defensive genes. We expanded the original sentence “For this reason, several well determined defensive genes have been cloned and studied in detail” as “For this reason, various kinds of well determined defensive genes have been cloned and studied in detail (Supplementary Table 5), such as lysins (fetidin, lysenin and coelomic cytolytic factor, i.e. CCF), antimicrobial proteins (lumbricin I, LBP/BPI, lysozyme), Toll-like receptors functioning in innate immune response against pathogens (mccTLR and sccTLR), proteins involved in response to oxidative stress (SOD, CAT and CRT) or metal stress (PCS), detoxification proteins (GST and CYP450) and heat shock protein (HSP70)”. We also added description for defensive genes in Introduction as “Last, we systematically profiled copies of several categories of well determined defensive genes in the genome, including lysins, antimicrobial proteins, Toll-like receptors, proteins involved in response to oxidative stress or metal stress, detoxification proteins and heat shock protein, and found their expansion trend in the genome as well as high diversity in the population of *A. corticis*”.

It makes sense that earthworms would be exposed to an array of microorganisms in the soil. However, it is not clear to me that pathogens are more common or diverse in soil than they are in, say, marine or freshwater sediments. It is also not clear to me that *Amyntas corticis* is unusual in its defensive ability in comparison to other earthworms. I certainly think it is interesting to study these defensive genes/proteins in *Amyntas*, but I think the authors should be more circumspect about what their findings might mean. It is possible that invasive earthworm species have more copies, and more variation, in their defensive genes than other earthworm species do, but it is also possible that ALL earthworms have more copies and more variation in their defensive genes than, say, leeches or *Capitella* or other spiralian animals have. Ideally, the authors would be able to compare their findings to other earthworm genomes. Unfortunately, though substantial work has been done on *Eisenia fetida*, that genome does not seem to be annotated. In short, I think we are too early in the game to be able to attribute invasive ability to expansions or higher levels of variation in defensive gene families.

Response: We appreciate the reviewer’s helpful suggestion and compared genomes of *Amyntas corticis*, *Eisenia andreii* and its close neighbor species *Eisenia fetida*^{1,6}. First, we evaluated ploidy for two earthworms in the genus of *Eisenia*. However, raw reads sequenced for *Eisenia andreii* were not released on any public platform, thus we only evaluated ploidy for *Eisenia fetida*. We found *Eisenia fetida* is diploid though the same analysis method. Given that *Eisenia fetida* is close neighbor species of *Eisenia*

andrei, it can be inferred that *Eisenia andrei* is also diploid with high probability. Then, we compared enriched functions of expanded gene families among these three earthworms and found counts of enriched functions associated with defenses, including response to stress, immune system process and homeostatic process, in *Amyntas corticis*, *Eisenia andrei* and *Eisenia fetida* were 131, 121, and 42, respectively. Also, we found enrichment levels of functions associated with response to stress (median odds ratio: 5.4 vs. 3.5, Wilcoxon rank sum test, p-value = 0.002) and immune system process (median odds ratio: 7.6 vs. 2.4, Wilcoxon rank sum test, p-value = 2e-12) in *Amyntas corticis* were significantly higher than those in *Eisenia fetida*, meanwhile enrichment levels of functions associated with homeostatic process (median odds ratio: 8.4 vs. 6.6, Wilcoxon rank sum test, p-value = 0.029) in *Amyntas corticis* were significantly higher than those in *Eisenia andrei*. Taken together, *Amyntas corticis* is not only triploid but also with more expanded gene families associated with defenses compare to other earthworms in the genus of *Eisenia*, implying its adaption ability was enhanced by both of these two characters. Of course, as is pointed by the reviewer, it is too early in the game to be able to attribute invasive ability to expansions or higher levels of variation in defensive gene families. Thus, we weakened the opinion by changing the original sentence “Profiling of molecular mechanisms coded in the genome of *A. corticis* helps to better understand the basis of global distribution and stress defensive ability of this species” to “Profiling of molecular mechanisms coded in the genome of *A. corticis* helps to better understand features associated with adaption ability of this species”.

I accept the first part of this sentence (*A. corticis* is different from *Helobdella* and *Capitella*), but not the second part, for the reasons discussed above.

Response: We appreciate the reviewer’s comment and removed the second part of this sentence from the manuscript.

Only abundance? Or changes in diversity/abundance of particular groups of gut bacteria?

Response: We appreciate the reviewer’s question and changed the original word “abundance” to “diversity as well as abundance”.

I do not think it matters much for the analyses presented here for *Amyntas*, but this is a rather bizarre phylogeny for these species.

Response: We checked the species tree displayed in the database of TreeFam (<http://www.treefam.org/browse#tabview=tab2>) and found most of partitions of our constructed tree are the same with it except for the root of flatworm lineage. However, the position of flatworm lineage is also not explicit in the species tree of TreeFam.

This is somewhat confusing, because “evolutionary rate” usually refers to rate of nucleotide or amino acid substitution, but the authors here are using it to mean the rate of growth/duplication of gene families. Perhaps another term should be used?

Response: We appreciate the reviewer’s comment and suggestion and we changed

the original phase “significantly accelerated evolutionary rate” to “significantly accelerated rates of gene duplication and loss”.

Again, this is unclear. Is it gut tissue only, or all body tissues except the gut?

Response: We appreciate the reviewer’s comment and changed the original phase “gut removed body” to “body tissues except the gut”.

Why not just write *Amyntas* instead of “earthworm” on the figure? They could also list the genus name of the python for consistency. The authors could even color code the word *Amyntas* or make it bold so that it is easier to see.

Response: We appreciate the reviewer’s helpful suggestions. We changed the original word “Earthworm” in related figures to “*Amyntas*”. We used the genus name “*Python*” in related figures to represent the species of “*Python bivittatus*”. Last, we made the word “*Amyntas*” bold.

References

- 1 Zwarycz, A. S., Nossa, C. W., Putnam, N. H. & Ryan, J. F. Timing and Scope of Genomic Expansion within Annelida: Evidence from Homeoboxes in the Genome of the Earthworm *Eisenia fetida*. *Genome biology and evolution* **8**, 271-281, doi:10.1093/gbe/evv243 (2015).
- 2 Hegarty, M. J. & Hiscock, S. J. Genomic clues to the evolutionary success of polyploid plants. *Curr Biol* **18**, R435-R444, doi:10.1016/j.cub.2008.03.043 (2008).
- 3 Finigan, P., Tanurdzic, M. & Martienssen, R. A. in *Polyploidy and genome evolution* 57-76 (Springer, 2012).
- 4 Sailer, C., Schmid, B. & Grossniklaus, U. Apomixis Allows the Transgenerational Fixation of Phenotypes in Hybrid Plants. *Curr Biol* **26**, 331-337, doi:10.1016/j.cub.2015.12.045 (2016).
- 5 Liu, X., Sun, Z., Chong, W., Sun, Z. & He, C. Growth and stress responses of the earthworm *Eisenia fetida* to *Escherichia coli* O157:H7 in an artificial soil. *Microb Pathog* **46**, 266-272, doi:10.1016/j.micpath.2009.02.001 (2009).
- 6 Wang, X., Chang, L. & Sun, Z. Differential expression of genes in the earthworm *Eisenia fetida* following exposure to *Escherichia coli* O157:H7. *Dev Comp Immunol* **35**, 525-529, doi:10.1016/j.dci.2010.12.014 (2011).
- 7 Wang, X., Chang, L., Sun, Z. & Zhang, Y. Comparative proteomic analysis of differentially expressed proteins in the earthworm *Eisenia fetida* during *Escherichia coli* O157:H7 stress. *J Proteome Res* **9**, 6547-6560, doi:10.1021/pr1007398 (2010).
- 8 Wang, X., Li, X. & Sun, Z. iTRAQ-based quantitative proteomic analysis of the earthworm *Eisenia fetida* response to *Escherichia coli* O157:H7. *Ecotoxicol Environ Saf* **160**, 60-66, doi:10.1016/j.ecoenv.2018.05.007 (2018).
- 9 Zhang, Y., Wang, G. , Wu, Y. , Zhao, H. , Zhang, Y. and Sun, Z. PCR-DGGE

- analysis of earthworm gut bacteria diversity in stress of *Escherichia coli* O157:H7. *Advances in Bioscience and Biotechnology* **4**, 437-441 (2013).
- 10 Weiss, C. L., Pais, M., Cano, L. M., Kamoun, S. & Burbano, H. A. nQuire: a statistical framework for ploidy estimation using next generation sequencing. *BMC Bioinformatics* **19**, 122, doi:10.1186/s12859-018-2128-z (2018).
- 11 Tatusov, R. L., Galperin, M. Y., Natale, D. A. & Koonin, E. V. The COG database: a tool for genome-scale analysis of protein functions and evolution. *Nucleic Acids Res* **28**, 33-36, doi:10.1093/nar/28.1.33 (2000).
- 12 Galperin, M. Y., Kristensen, D. M., Makarova, K. S., Wolf, Y. I. & Koonin, E. V. Microbial genome analysis: the COG approach. *Brief Bioinform* **20**, 1063-1070, doi:10.1093/bib/bbx117 (2019).
- 13 Shao, Y. *et al.* Genome and single-cell RNA-sequencing of the earthworm *Eisenia andrei* identifies cellular mechanisms underlying regeneration. *Nat Commun* **11**, 2656, doi:10.1038/s41467-020-16454-8 (2020).
- 14 Paul, S. *et al.* Data on genome annotation and analysis of earthworm *Eisenia fetida*. *Data Brief* **20**, 525-534, doi:10.1016/j.dib.2018.08.067 (2018).
- 15 National Genomics Data Center, M. & Partners. Database Resources of the National Genomics Data Center in 2020. *Nucleic Acids Res* **48**, D24-D33, doi:10.1093/nar/gkz913 (2020).

REVIEWERS' COMMENTS:

Reviewer #1 (Remarks to the Author):

I have a few points:

1. There is no reference to Supplemental Table 6 in the main manuscript.
2. I ran BUSCO on the Eukaryota database using Gvolante web interface. The results are here: <https://gvolante.riken.jp/script/result.cgi?202010310443-CMSU6WNYFRP2BT3V>
The 2 measures I requested "Average number of orthologs per core genes" and "% of detected core genes that have more than 1 ortholog" are reported here as 1.25 and 22.22 respectively. It's important that these numbers be reported. The authors used the Metazoa database rather than the Eukaryota database, so either the completeness numbers from this Gvolante run should be used or the authors can rerun via the web interface and specify the Metazoa database. I would argue that these numbers suggest that haplotigs are not a problem with this assembly.
3. As pointed out by the other reviewers, the position of Platyhelminthes in Figure 3B is strange. This weird placement of Platyhelminthes in this tree may cause readers to doubt the integrity of the rest of the analyses in this paper. I suggest removing this tree and the subsequent analyses. If the authors would like to provide a phylogenetic tree, I would recommend using an Orthofinder approach (see for example: <https://www.sciencedirect.com/science/article/abs/pii/S096098222030587X>).

Reviewer #2 (Remarks to the Author):

I think my concerns have been mainly addressed in the revised MS, but for the comment concerning the results of the acute ecotoxicological test with E.coli O157, the authors just give some inadequate explanations and prefer to retain this part in the MS. I still suggest to remove this part from the MS, because it has nothing to do with adaptability of a species in the sense of adaptive evolution. There are also some corrections should be made in "Supplementary Fig. 2 COI barcode phylogeny for Amynthus": 1)"The phylogenetic tree" is just a gene tree; 2)Amynthus homochaetus is a synonym of Amynthus corticis; 3)Amynthus dandongensis is not yet formally published, so it should be "Amynthus sp"; 4)Amynthus pingi is synonym of Amynthus canosus.

Reviewer #3 (Remarks to the Author):

The authors seem to have addressed all of my concerns reasonably well. In particular, I appreciate the addition of detailed comparisons with Eisenia and the information about the specimen from which the genomic data were collected. My only additional request along these lines is that they clearly label the COI barcode sequence from the specimen whose genome was sequenced in this study on the tree in Supplementary Figure 2. Right now, there are simply numbers and a few species names at the tips of the trees, and it is not clear which tip (number) represents the COI barcode extracted from the A. corticis genome discussed in this paper. It should be clearly labeled, or an arrow could be used to point to it.

I have only a few remaining suggestions that I think will improve the manuscript further.

(line numbers refer to the marked-up PDF)

Line 33: "Earthworms (Annelida: Crassiditellata) are widely distributed around the world due to their ancient origination as well as adaptation and invasion after introduced in new habitats over the past few centuries." would be better as "...as adaptation and invasion after introduction into new habitats over the past few centuries"

Line 45: "...in the body of *A. corticis* as well as 16S rDNA" may be better as "...in the body of *A. corticis* and 16S rDNA..."

Line 101: "and greater adaptation of *A. corticis*" -- A very minor point, but the authors said in their response that they deleted "greater" here, but they did not actually do so. This is not a big problem, but it retains the ambiguity of what *Amyntas*'s adaptation is greater than. I am now convinced that *Amyntas* is doing different things than *E. fetida*, but that is not clear yet at this point in the manuscript.

Line 144: "Previous studies have demonstrated that expansion of sodium-potassium pump alpha-subunit and several other gene families in leech happened at the same time of its transition from marine to freshwater habitats and might play important roles in diversification of annelid into freshwater habitat" -- Might be better as "as its" (rather than "of its") and "of annelids into freshwater habitats" (rather than "of annelid into freshwater habitat").

Line 148: "To reveal how expansion of gene families affected earthworm *A. corticis*, we constructed gene families for *A. corticis*, invertebrates, including annelids, flatworms, roundworms, mollusc and insect, and vertebrates based on the database of TreeFam." -- This is a somewhat awkward sentence. Perhaps it would be better as "To reveal how expansion of gene families affected earthworm *A. corticis*, we constructed gene families for *A. corticis*, and several other invertebrates (including annelids, flatworms, roundworms, molluscs and insects) and vertebrates based on the TreeFam database."

December 02, 2020

RE: Revision of manuscript COMMSBIO-20-1340A

Wang et al.,

Genome reveals molecular mechanisms behind global distribution of *Amyntas corticis*

Dear reviewers,

We thank the Communications Biology for their careful comments. Those comments are valuable and helpful for revising and improving our manuscript, as well as the important guiding significance to our researches. We have studied comments carefully and have made correction which we hope meet with approval. Revised portion are marked in red in the revised manuscript. The responds to the reviewer's comments are as following:

REVIEWERS' COMMENTS:

Reviewer #1 (Remarks to the Author):

I have a few points:

1. There is no reference to Supplemental Table 6 in the main manuscript.

Response: We appreciate the reviewer's reminder and added the reference to Supplemental Table 6 in the newly revised manuscript.

2. I ran BUSCO on the Eukaryota database using Gvolante web interface. The results are

here:<https://gvolante.riken.jp/script/result.cgi?202010310443-CMSU6WNYFRP2BT3>

V

The 2 measures I requested "Average number of orthologs per core genes" and "% of

detected core genes that have more than 1 ortholog" are reported here as 1.25 and 22.22 respectively. It's important that these numbers be reported. The authors used the Metazoa database rather than the Eukaryota database, so either the completeness numbers from this Gvolante run should be used or the authors can rerun via the web interface and specify the Metazoa database. I would argue that these numbers suggest that haplotigs are not a problem with this assembly.

Response: We appreciate the reviewer's suggestion and rerun gVolante via the web interface using Metazoa database. "Average number of orthologs per core genes" and "% of detected core genes that have more than 1 ortholog" were 1.3 and 24.78 respectively. These 2 measures were reported in the newly revised manuscript, while the method of gVolante was added in Method section.

3. As pointed out by the other reviewers, the position of Platyhelminthes in Figure 3B is strange. This weird placement of Platyhelminthes in this tree may cause readers to doubt the integrity of the rest of the analyses in this paper. I suggest removing this tree and the subsequent analyses. If the authors would like to provide a phylogenetic tree, I would recommend using an Orthofinder approach (see for example: <https://www.sciencedirect.com/science/article/abs/pii/S096098222030587X>).

Response: We appreciate the reviewer's suggestion and removed this phylogenetic tree and related text from the newly revised manuscript.

Reviewer #2 (Remarks to the Author):

I think my concerns have been mainly addressed in the revised MS, but for the comment concerning the results of the acute ecotoxicological test with E.coli O157, the authors just give some inadequate explanations and prefer to retain this part in the MS. I still suggest to remove this part from the MS, because it has nothing to do with adaptability of a species in the sense of adaptive evolution. There are also some corrections should be made in "Supplementary Fig. 2 COI barcode phylogeny for Amynthes": 1)"The phylogenetic tree" is just a gene tree; 2)Amynthes homochaetus is

a synonym of *Amyntas corticis*; 3) *Amyntas dandongensis* is not yet formally published, so it should be "*Amyntas sp.*"; 4) *Amyntas pingi* is synonym of *Amyntas canosus*.

Response: We appreciate the reviewer's comment on results of the acute ecotoxicological test with *E. coli* O157:H7 and totally agree the reviewer's opinion that it was not fully representative of the adaptability of *Amyntas corticis*. For the consideration of the completeness of this research, we decided to retain this part of results without overinterpretation on adaptation. As the reviewer's suggestions, we had changed several corrections which should be made in Supplementary Fig. 2: 1) changed "the phylogenetic tree" to "the gene tree"; 2) changed "*Amyntas homochaetus*" to "*Amyntas corticis*"; 3) changed "*Amyntas dandongensis*" to "*Amyntas sp.*"; 4) changed "*Amyntas pingi*" to "*Amyntas canosus*".

Reviewer #3 (Remarks to the Author):

The authors seem to have addressed all of my concerns reasonably well. In particular, I appreciate the addition of detailed comparisons with *Eisenia* and the information about the specimen from which the genomic data were collected. My only additional request along these lines is that they clearly label the COI barcode sequence from the specimen whose genome was sequenced in this study on the tree in Supplementary Figure 2. Right now, there are simply numbers and a few species names at the tips of the trees, and it is not clear which tip (number) represents the COI barcode extracted from the *A. corticis* genome discussed in this paper. It should be clearly labeled, or an arrow could be used to point to it.

Response: We appreciate the reviewer's suggestion and used an arrow to point to the tip representing the COI barcode sequence from the specimen whose genome was sequenced in this study on the tree in Supplementary Figure 2.

I have only a few remaining suggestions that I think will improve the manuscript further.

(line numbers refer to the marked-up PDF)

Line 33: “Earthworms (Annelida: Crassiditellata) are widely distributed around the world due to their ancient origination as well as adaptation and invasion after introduced in new habitats over the past few centuries.” would be better as “...as adaptation and invasion after introduction into new habitats over the past few centuries”

Response: Following the reviewer’s suggestion, we had changed “Earthworms (Annelida: Crassiditellata) are widely distributed around the world due to their ancient origination as well as adaptation and invasion after introduced in new habitats over the past few centuries.” to “Earthworms (Annelida: Crassiditellata) are widely distributed around the world due to their ancient origination as well as adaptation and invasion after introduction into new habitats over the past few centuries”.

Line 45: “...in the body of A. corticis as well as 16S rDNA” may be better as “...in the body of A. corticis and 16S rDNA...”

Response: Following the reviewer’s suggestion, we had changed “...in the body of A. corticis as well as 16S rDNA” to “...in the body of A. corticis and 16S rDNA...”.

Line 101: “and greater adaptation of A. corticis” -- A very minor point, but the authors said in their response that they deleted “greater” here, but they did not actually do so. This is not a big problem, but it retains the ambiguity of what Amynthis’s adaptation is greater than. I am now convinced that Amynthis is doing different things than E. fetida, but that is not clear yet at this point in the manuscript.

Response: We had deleted “greater” here now.

Line 144: “Previous studies have demonstrated that expansion of sodium-potassium pump alpha-subunit and several other gene families in leech happened at the same time of its transition from marine to freshwater habitats and might play important roles in diversification of annelid into freshwater habitat” -- Might be better as “as its” (rather than “of its”) and “of annelids into freshwater habitats” (rather than “of annelid into freshwater habitat”).

Response: We appreciate the reviewer's suggestion and changed "Previous studies have demonstrated that expansion of sodium-potassium pump alpha-subunit and several other gene families in leech happened at the same time of its transition from marine to freshwater habitats and might play important roles in diversification of annelid into freshwater habitat" to "Previous studies have demonstrated that expansion of sodium-potassium pump alpha-subunit and several other gene families in leech happened at the same time as its transition from marine to freshwater habitats and might play important roles in diversification of annelids into freshwater habitats".

Line 148: "To reveal how expansion of gene families affected earthworm *A. corticis*, we constructed gene families for *A. corticis*, invertebrates, including annelids, flatworms, roundworms, mollusc and insect, and vertebrates based on the database of TreeFam." – This is a somewhat awkward sentence. Perhaps it would be better as "To reveal how expansion of gene families affected earthworm *A. corticis*, we constructed gene families for *A. corticis*, and several other invertebrates (including annelids, flatworms, roundworms, molluscs and insects) and vertebrates based on the TreeFam database."

Response: We appreciate the reviewer's suggestion and changed "To reveal how expansion of gene families affected earthworm *A. corticis*, we constructed gene families for *A. corticis*, invertebrates, including annelids, flatworms, roundworms, mollusc and insect, and vertebrates based on the database of TreeFam" to "To reveal how expansion of gene families affected earthworm *A. corticis*, we constructed gene families for *A. corticis*, several other invertebrates (including annelids, flatworms, roundworms, molluscs and insects) and vertebrates based on the TreeFam database".